# The OTUD6B-LIN28B-MYC axis determines the proliferative state in multiple myeloma

Carmen Paulmann[1,2,†], Ria Spallek[1,2,†], Oleksandra Karpiuk[1,2], Michael Heider[1,2], Isabell Schäffer[1,2], Jana Zecha[3], Susan Klaeger[3], Michaela Walzik[1,2], Rupert Öllinger[2,4,5], Thomas Engleitner[2,4,5], Matthias Wirth[6], Ulrich Keller[6,7,8], Jan Krönke[6,7], Martina Rudelius[9], Susanne Kossatz[2,10], Roland Rad[2,4,5,7], Bernhard Kuster[3,7] & Florian Bassermann[1,2,7,*]

## Abstract

Deubiquitylases (DUBs) are therapeutically amenable components of the ubiquitin machinery that stabilize substrate proteins. Their inhibition can destabilize oncoproteins that may otherwise be undruggable. Here, we screened for DUB vulnerabilities in multiple myeloma, an incurable malignancy with dependency on the ubiquitin proteasome system and identified OTUD6B as an oncogene that drives the G1/S-transition. LIN28B, a suppressor of microRNA biogenesis, is specified as a bona fide cell cycle-specific substrate of OTUD6B. Stabilization of LIN28B drives MYC expression at G1/S, which in turn allows for rapid S-phase entry. Silencing OTUD6B or LIN28B inhibits multiple myeloma outgrowth *in vivo* and high OTUD6B expression evolves in patients that progress to symptomatic multiple myeloma and results in an adverse outcome of the disease. Thus, we link proteolytic ubiquitylation with post-transcriptional regulation and nominate OTUD6B as a potential mediator of the MGUS-multiple myeloma transition, a central regulator of MYC, and an actionable vulnerability in multiple myeloma and other tumors with an activated OTUD6B-LIN28B axis.

**Keywords** cell cycle; deubiquitylases; multiple myloma; RNA binding proteins; ubiquitin

**Subject Categories** Cancer; Cell Cycle; Post-translational Modifications & Proteolysis

**The EMBO Journal (2022) 41: e110871**

## Introduction

Multiple myeloma (MM) is the second most common hematological malignancy with an adverse outcome reflected by a median survival of ~5 years (Mikhael *et al*, 2019). The pathophysiology of MM is poorly understood, but high response rates to therapies targeting the ubiquitin-proteasome-system (UPS) such as the proteasome inhibitors bortezomib or carfilzomib, suggest that aberrant functions of the UPS drive and maintain the disease (Kumar *et al*, 2017; Guerrero-Garcia *et al*, 2018). MM mandatorily evolves from a pre-malignant state termed monoclonal gammopathy of undetermined significance (MGUS). These cells are genetically similar to MM but fail to enter S-phase of the cell cycle, a prerequisite for the progression towards symptomatic MM (Chng *et al*, 2011; van Nieuwenhuijzen *et al*, 2018). This disease thus provides the ideal opportunity to study cancer relevant vulnerabilities within the ubiquitin system and among cell cycle checkpoints, not only to accommodate for the high clinical need for novel therapeutic strategies in MM, but also to unravel common oncogenic mechanisms that can serve as actionable targets in a broader oncology context.

As a central post-translational means of protein homeostasis, the UPS regulates all dimensions of cellular life and cancer cells exploit this machinery for their evolvement, maintenance and evolution (Bassermann *et al*, 2013; Oh *et al*, 2018). Ubiquitylation can be reversed by deubiquitylases (DUBs; Komander *et al*, 2009; Cheng *et al*, 2019). These enzymes have recently gained attention as drug-gable proteases with actionable catalytic sites that can give rise to enhanced expression and activity of cancer relevant proteins (Harri-gan *et al*, 2018). Indeed, although most of the ∼100 mammalian DUBs remain to be linked to their substrates, their biological

---

1   Department of Medicine III, Klinikum Rechts der Isar, Technical University of Munich, Munich, Germany
2   TranslaTUM, Center for Translational Cancer Research, Technical University of Munich, Munich, Germany
3   Chair of Proteomics and Bioanalytics, Technical University of Munich, Freising, Germany
4   Department of Medicine II, Klinikum Rechts der Isar, Technical University of Munich, Munich, Germany
5   Institute of Molecular Oncology and Functional Genomics, Technical University of Munich, Munich, Germany
6   Department of Hematology, Oncology and Cancer Immunology, Campus Benjamin Franklin Charité – Universitätsmedizin Berlin, corporate member of Freie Universität Berlin and Humboldt-Universität zu Berlin, Berlin, Germany
7   Deutsches Konsortium für Translationale Krebsforschung (DKTK), Heidelberg, Germany
8   Max-Delbrück-Center for Molecular Medicine, Berlin, Germany
9   Institute of Pathology, Ludwig Maximilians University, Munich, Germany
10  Department of Nuclear Medicine, Klinikum Rechts der Isar, Technical University of Munich, Munich, Germany
    *Corresponding author. Tel: +49 89 4140 4111; E-mail: florian.bassermann@tum.de
    †These authors contributed equally to this work

activities, or both, different characterized members have well-defined oncogenic functions, implicating an important role for this protein family in cancer (Harrigan *et al*, 2018).

Post-transcriptional regulation of key oncogenic pathways by RNA-binding proteins (RBP) is another emerging feature in cancer (Kang *et al*, 2020). LIN28B is such an RBP that functions as a master regulator of lethal 7 (let-7) microRNAs. As such, it prevents Dicer and Drosha from generating mature let-7 miRNA resulting in a de-repression and up-regulation of important cancer relevant let-7 targets, including MYC, RAS, VEGF, PDK1 and E2F (Heo *et al*, 2009; Viswanathan *et al*, 2009; Balzeau *et al*, 2017). LIN28B exerts highest expression in embryonic stem cells reflecting its important physiological function as a pluripotency factor, while its expression is silenced in differentiated somatic cells (Shyh-Chang & Daley, 2013; Segalla *et al*, 2015; Zhang *et al*, 2016). However, LIN28B expression becomes re-established in certain tumors, including MM. This expression pattern distinguishes LIN28B as an attractive therapeutic target, but pharmacological strategies to target LIN28B and upstream regulators have remained largely elusive (Viswanathan *et al*, 2008; Balzeau *et al*, 2017; Manier *et al*, 2017; Ustianenko *et al*, 2018).

We here set out to identify cancer relevant vulnerabilities among the family of DUBs using MM as a relevant model disease. These studies uncovered OTUD6B as a central dependency in MM, whose activity directly impinges on the RBP LIN28B in order to drive a MYC-dependent cell cycle activatory program, that links post-translational activity of the UPS with post-transcriptional gene regulation and may contribute to both the conversion of MGUS to MM and the maintenance of MM.

## Results

### Identification of OTUD6B as a dependency in MM that promotes the G1/S cell cycle transition

To identify DUBs that promote and maintain MM and may serve as novel therapeutic targets in this disease, we performed a CRISPR/Cas9-based screen targeting all DUBs in human MM cells (Figs 1A and EV1A, Appendix Table S1). Highest scoring hits (Appendix Table S1) were then subjected to a selection process based on novelty, essentiality, prognostic impact in MM and MM cell competition experiments (Appendix Fig S1). This strategy eventually identified

OTUD6B as the most promising candidate that we show to promote proliferation in different MM cells (Figs 1B and EV1A–E). Specificity was further substantiated by the re-expression of OTUD6B in OTUD6B silenced MM cells, that reversed the phenotype (Fig EV1F and G).

OTUD6B is a largely uncharacterized DUB of the cysteine protease family (Harrigan *et al*, 2018). Mice homozygous for Otud6b knock-out alleles are sub-viable and biallegic pathogenic variants in *OTUD6B* associate with an intellectual disability syndrome and dysmorphic features in humans suggesting a role in embryonic development (Santiago-Sim *et al*, 2017; Straniero *et al*, 2018).

Because our initial experiments suggested a role for OTUD6B in cell proliferation, we next studied OUTD6B within the context of cell cycle regulation. Indeed, OTUD6B knockdown resulted in a strong decrease in the S-phase population and a significant increase in cells at the G1/S transition in several MM cell lines with and without MAF translocations/deregulations (Fig 1C; Appendix Fig S2A and B). This phenotype arose from a DNA-damage independent failure of OTUD6B-depleted MM cells to enter S-phase (Fig 1D and E), which eventually resulted in apoptosis induction (Appendix Fig S2C). OTUD6B depletion also led to a G1/S arrest and reduced proliferation in cancer cells of epithelial origin such as A549 lung adenocarcinoma cells, suggesting a more general role of OTUD6B in promoting cell cycle progression (Fig EV2A–C). In line with these observations, the catalytic activity of OTUD6B increased during G1-phase and peaked at G1/S, thus further suggesting a specific function of OTUD6B in promoting the G1/S transition (Figs 1F, and EV2D and E). OTUD6B protein expression levels did not change throughout the cell cycle indicating other means by which OTUD6B becomes activated in a cell cycle-dependent manner (Fig 1F).

### LIN28B is a direct deubiquitylation substrate of OTUD6B

In search for substrates of OTUD6B that become deubiquitylated, we performed proteome-wide affinity- (Tag-based purification) and non-affinity-based (Bio-ID-purification) screens combined with mass spectrometric analyses (Fig EV3A–C). Cross-validation of both approaches identified the RNA-binding protein LIN28B as the most promising substrate candidate (Fig 2A).

First, we confirmed specific bi-directional binding between LIN28B and OTUD6B using different members of the OTU DUB family as controls (Fig EV3D and E). LIN28B only interacted with

**Figure 1. Identification of OTUD6B as a dependency in MM that promotes the G1/S cell cycle transition.**

A   Averaged sgRNA representation of a DUB CRISPR drop out screen in MM1.S cells clustered per gene. The ratio of normalized sgRNA read-counts on day 14 versus day 0 was determined for each sgRNA and the average fold change blotted per gene (except for non-targeting controls). Lines represent the median. Essential genes included as controls are: POLR2, PRPF8, RPL32, RPA3, RPL8, RPS19. OTUD6B is marked in red.

B   Cell proliferation analysis of different MM cell lines, expressing shRNAs targeting OTUD6B as counted by trypan blue exclusion method. Numbers are depicted as fold of shCtrl on day 8 after infection ($n = 3$ independent experiments).

C   Cell cycle analyses of JJN3, MM1.S, RPMI8226, L363 and U266 cells expressing the indicated shRNAs by BrdU/PI flow cytometry ($n = 3$ independent experiments).

D   Flow cytometric analysis of PI stained MM1.S cells transduced with the indicated shRNAs before and after release from G1/S block. Left: Exemplary FACS-plots. Right: Quantification of three independent experiments showing the percentage of cells in G1 phase.

E   Immunoblot analysis of cells described in (D), harvested at the indicated time points after G1/S release. Exemplary blot from three independent experiments.

F   DUB activity assay for OTUD6B in asynchronous, G1/S- or mitotically synchronized and released A549 cells using HA-ubiquitin vinyl-sulfone to isolate active forms of DUBs. Analysis was performed by immunoblotting using the indicated antibodies.

Data information: In (B–D), data are mean ± s.d. *$P < 0.05$; **$P < 0.01$; ***$P < 0.001$; by paired Student's *t*-test (B–D) corrected for multiple testing (B, D).
Source data are available online for this figure.

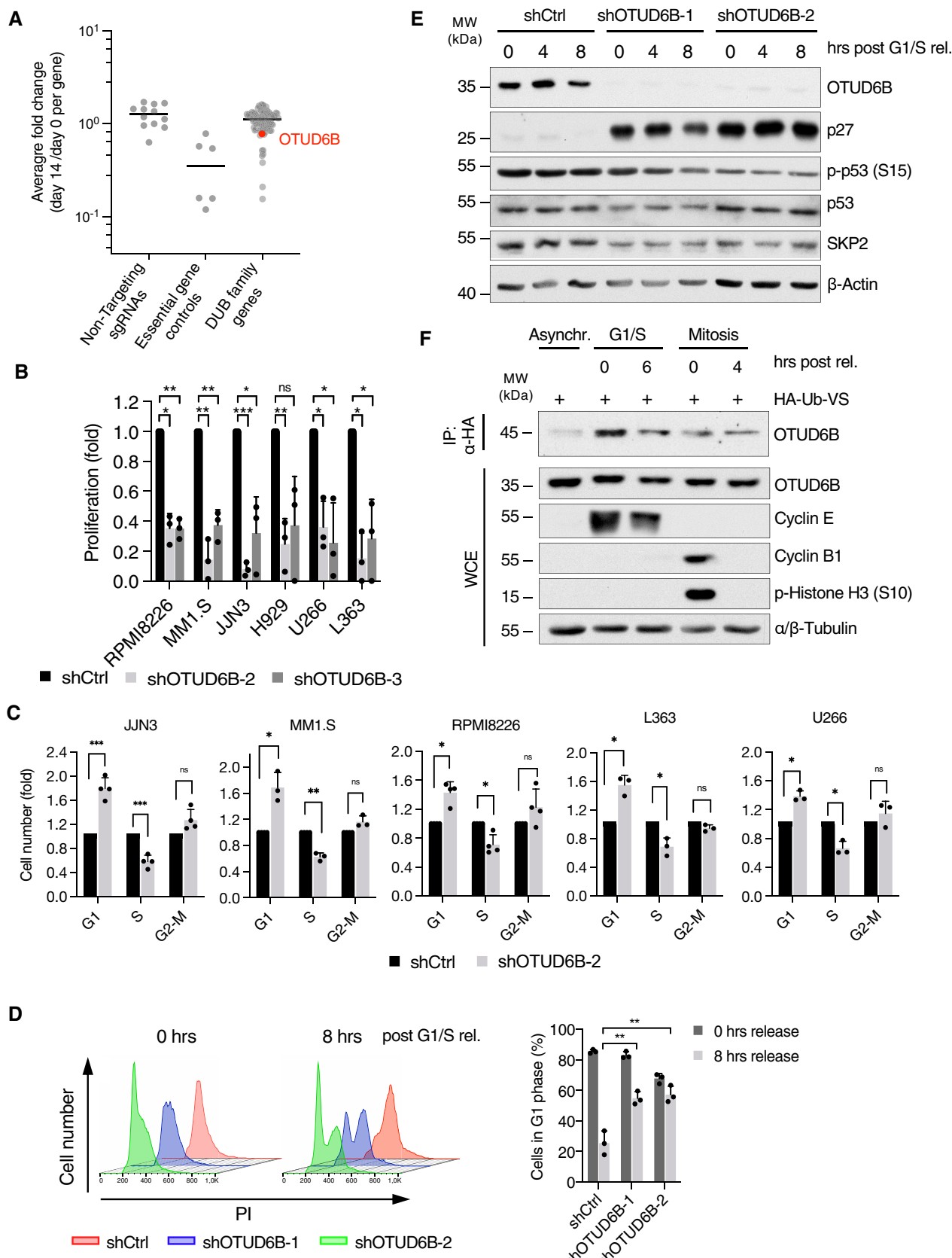

**Figure 1.**

isoform 1 of OTUD6B, suggesting binding within the N-terminus of this isoform (Fig EV3F). Further mapping located the OTUD6B-binding region to the cold shock domain of LIN28B (Appendix Fig S3).

We next addressed the imminent question, whether LIN28B is a direct deubiquitylation substrate of OTUD6B. Indeed, siRNA-mediated knockdown of OTUD6B substantially enhanced polyubiquitylation of LIN28B (Fig 2B), whereas forced expression of OTUD6B decreased LIN28B ubiquitylation (Figs 2C and EV4A). Specificity of this deubiquitylation event was ascertained by overexpression of the catalytic inactive OTUD6B mutant (C158A), which deubiquitylated LIN28B to a substantially smaller extent (Fig 2C).

We further thought to reconstitute this deubiquitylation reaction *in vitro*. To this end, we purified GST-OTUD6B from bacteria and confirmed direct binding thereof to LIN28B (Fig 2D). Next, GST-OTUD6B was inserted into a reaction with FLAG-purified ubiquitylated LIN28B from 293T cells. OTUD6B readily deubiquitylated LIN28B in this *in vitro* reconstituted system, suggesting that LIN28B is a direct substrate (Fig 2E). Moreover, using K48 linkage-specific ubiquitin moieties and a ubiquitin-K48 specific antibody, we found that OTUD6B specifically cleaves K48-branched ubiquitin chains from LIN28B that determine proteolytic substrate fates (Figs 2C and E, and EV4B).

Together, these data identify LIN28B as a direct K48-specific deubiquitylation substrate of OTUD6B.

### OTUD6B stabilizes LIN28B at the G1/S cell cycle transition point

We next investigated whether LIN28B is a G1/S-specific deubiquitylation substrate of OTUD6B. To this end, we compared binding of OTUD6B to LIN28B in G1/S synchronized MM1.S cells and cells released from the cell cycle block. These experiments revealed that binding of OTUD6B to LIN28B specifically occurs at G1/S (Fig 3A).

Having shown G1/S-specific binding and K48-specific deubiquitylation of LIN28B by OTUD6B, we tested to what extend OTUD6B influences LIN28B stability in a cell cycle-dependent manner. First, we analyzed whether LIN28B exerts a cell cycle-dependent expression profile under normal conditions in MM and epithelial lung cancer cells. Indeed, we found, that LIN28B expression peaks at G1/S and early S-phase, while its expression decreases in late S-phase and mitosis, before it becomes re-established in G1-phase, in line with the deubiquitylase activity profile of OTUD6B (Figs 3B and C and 1F; Appendix Fig S4A and B). Subsequently, we silenced OTUD6B in cells that were either left asynchronous or synchronized at G1/S and performed LIN28B half-life analyses. These studies demonstrated that OTUD6B loss results in a dramatically reduced half-life of LIN28B in G1/S cells, while this effect was largely absent in asynchronous cells (Fig 3D; Appendix Fig S4C and D). The expression of LIN28B could be re-established by the addition of the proteasome inhibitor MG132, indicating that LIN28B is indeed subjected to proteasomal degradation (Appendix Fig S4D).

Together, these data suggest that LIN28B is a cell cycle regulated protein with highest abundance at the G1/S transition and early S-phase and that OTUD6B specifically deubiquitylates LIN28B at the G1/S transition of the cell cycle to procure its stability.

### The LIN28B-OTUD6B axis is a vulnerability in MM that drives cell cycle progression *in vitro* and *in vivo*

We next explored the (patho-)physiological relevance of OTUD6B-mediated stabilization of LIN28B and analyzed its impact on cell cycle progression. After synchronization of MM cells at the G1/S restriction point using the CDK4/6 inhibitor palbociclib, OTUD6B- and LIN28B-depleted cells remained in G1 upon cell cycle release whereas control cells readily progressed to S- and G2/M-phases (Fig 4A and B). Importantly, induced ectopic expression of LIN28B rescued the G1/S cell cycle block evoked by OTUD6B depletion in MM cells, thus confirming that OTUD6B drives S-phase entry and cell proliferation via LIN28B (Fig 4C and D; Appendix Fig S4E).

Given the strong implication of the OTUD6B-LIN28B axis in cell cycle progression, we next performed *in vivo* xenograft experiments with human MM cells to further validate this nexus as a central vulnerability in MM. Indeed, both OTUD6B and LIN28B inactivation led to a significant reduction of MM tumor size, weight and volume and IHC analyses of the respective tumors showed an induction of apoptosis in both OTUD6B and LIN28B depleted tumors (Fig 4E–G).

These data distinguish the OTUD6B-LIN28B DUB-substrate pair as central dependency in MM cells *in vitro* and *in vivo*.

---

**Figure 2.  LIN28B is a direct deubiquitylation substrate of OTUD6B.**

A   Results of mass spectrometric (MS) based screening for OTUD6B substrates correlating the results from a FLAG- and a Bio-ID-purification. Intensities (LFQ) of co-immunoprecipitated proteins identified by MS were log2 transformed and the differences between OTUD6B and EV plotted against LFQ intensities of proteins identified in the OTUD6B sample. The red dotted line represents the cut off for two-fold enrichment in the OTUD6B sample compared with control. Proteins enriched more than two-fold in the BioID-proximity screen are highlighted in green. OTUD6B (bait) and LIN28B are depicted in red.

B   *In vivo* ubiquitylation analyses of LIN28B in HEK293T cells in which OTUD6B was silenced by siRNA. Cells were transfected with the indicated siRNAs and overexpression constructs, then treated with MG132 for 3 h. Lysis and IP was done under denaturing conditions followed by WB analysis.

C   *In vivo* ubiquitylation assay of LIN28B in OTUD6B and a catalytically inactive variant of OTUD6B (OTUD6B-C158A) overexpressing cells. HEK293T cells were transfected with indicated combinations of FLAG-LIN28B, HA-Ubiquitin, OTUD6B, OTUD6B-C158A and EV control and treated with MG132 for 3 h 24 h later. Denatured WCE were subjected to FLAG-IP. WCE and IP were analyzed by immunoblotting including a K48-specific ubiquitin antibody. Exemplary blots from three independent experiments are shown.

D   Pull-down of endogenous LIN28B from HEK293T WCE using bacterially purified GST-OTUD6B. Pulldowns and WCE were analyzed by immunoblot using the indicated antibodies.

E   Immunoblot analysis of an *in vitro* deubiquitylation assay using GST-purified OTUD6B from bacteria and FLAG-purified ubiquitylated LIN28B from HEK293T. FLAG-IP was performed under denaturing conditions from HEK293T WCEs expressing FLAG-LIN28B and HA-Ubiquitin. Purified FLAG-LIN28B was eluted from the beads and incubated with GST or GST-OTUD6B proteins followed by immunoblot analysis.

Data information: For (B, D, and E) exemplary blot of two independent experiments.
Source data are available online for this figure.

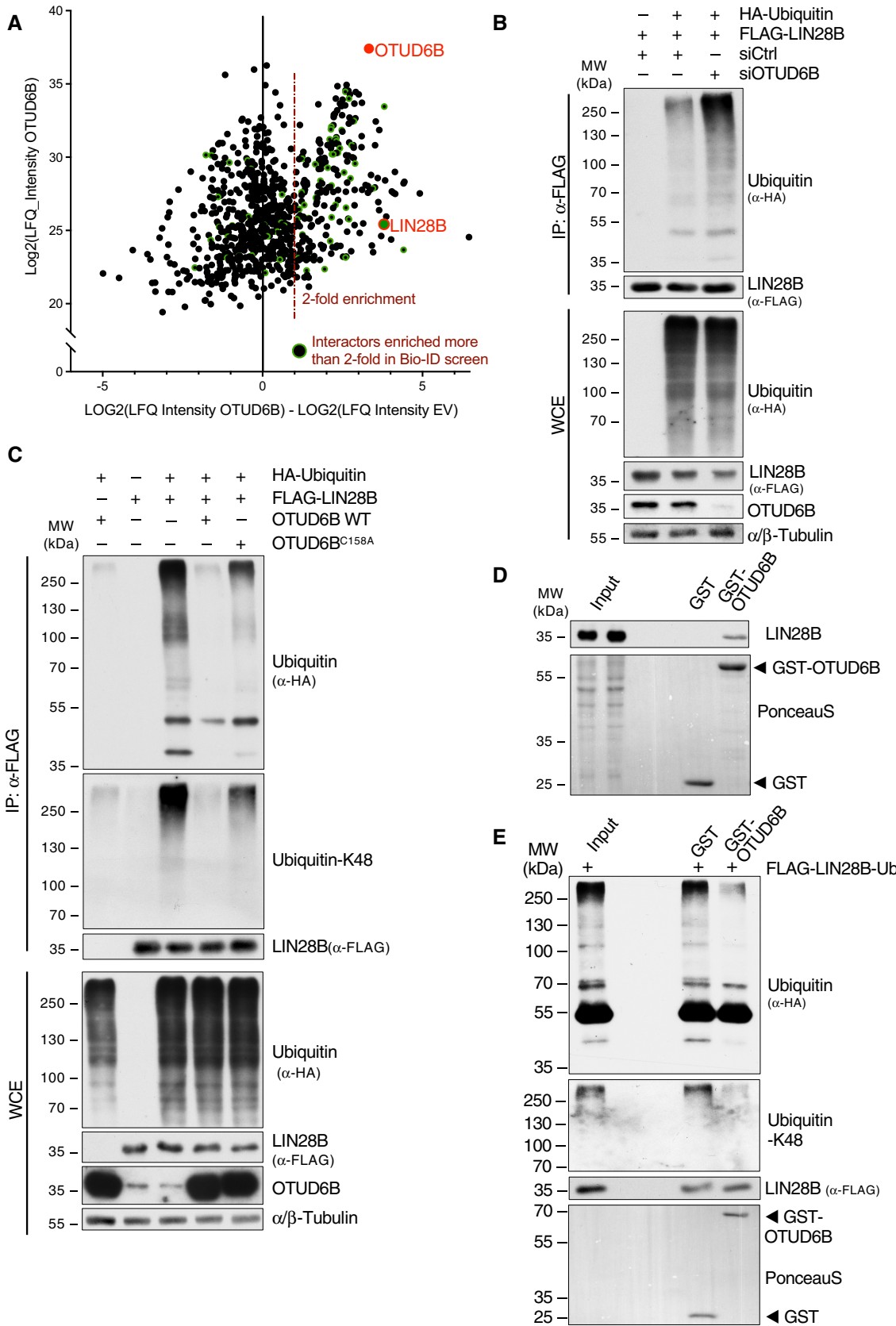

**Figure 2.**

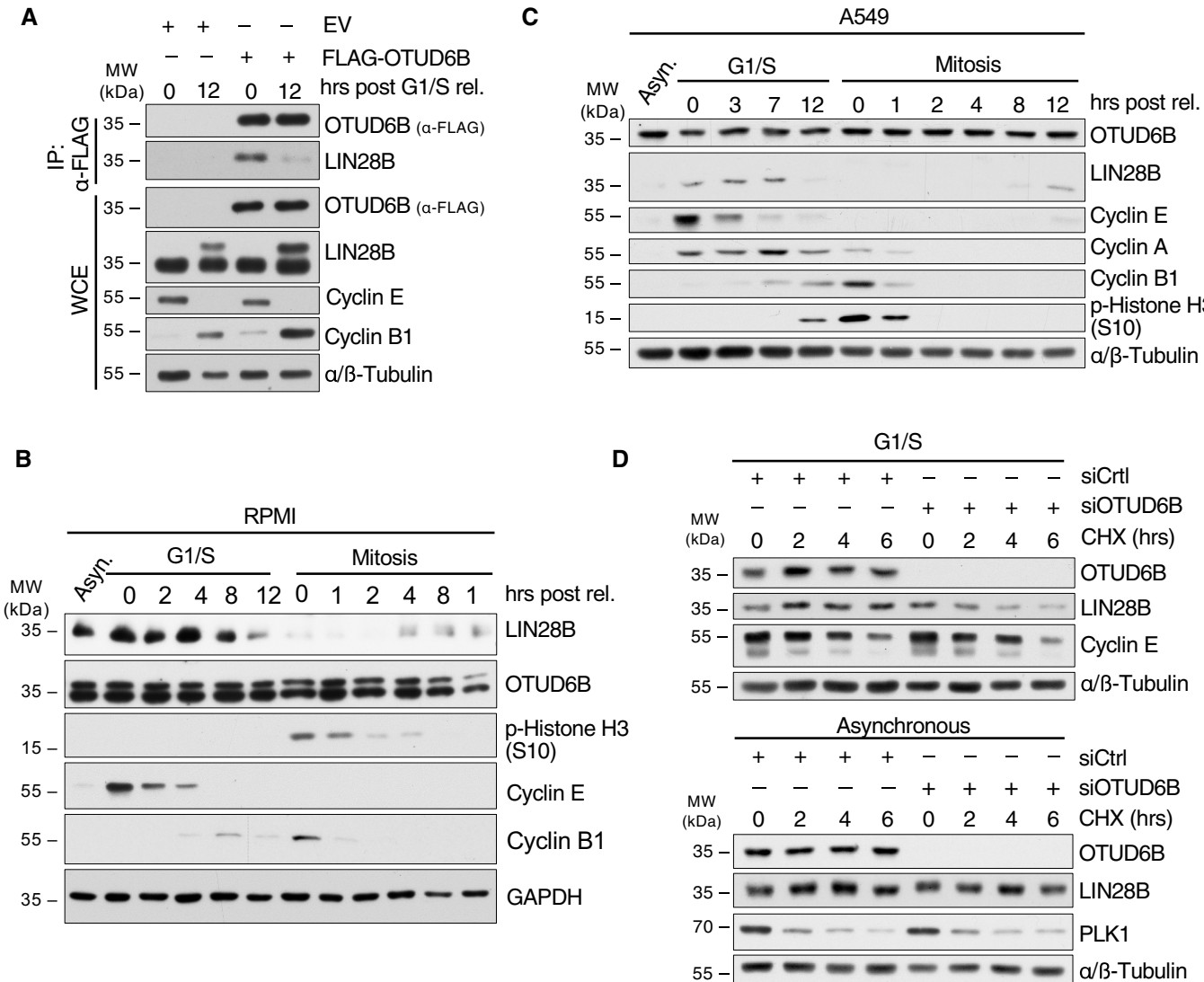

**Figure 3. OTUD6B stabilizes LIN28B at the G1/S cell cycle transition point.**

A    Co-IP of FLAG-OTUD6B with endogenous LIN28B from MM1.S cells synchronized in G1/S and 12 h post release. MM1.S cells stably expressing FLAG-OTUD6B or FLAG-EV were synchronized at G1/S and harvested at the indicated time points after release. IPs were performed and analyzed together with WCE by immunoblotting.

B, C  Immunoblot analysis of LIN28B and OTUD6B protein levels throughout the cell cycle. RPMI (B) and A549 (C) cells were synchronized at the G1/S transition by a double thymidine block or in mitosis by a sequential thymidine/nocodazole block. Synchronized cells were released, harvested at the indicated time points and analyzed by immunoblot using the indicated antibodies.

D    Immunoblot analysis of LIN28B protein half-life in G1/S-synchronized (upper panel) and asynchronous (lower panel) A549 cells upon OTUD6B depletion. Cells transfected with the respective siRNAs were synchronized at the G1/S transition or not and treated with cycloheximide (CHX) as indicated.

Data information: (A–D) All blots are representations of three independent experiments.
Source data are available online for this figure.

## OTUD6B activates MYC via LIN28B

To understand how the OTUD6B-LIN28B axis regulates the G1/S-transition, we performed RNA-Seq experiments in MM cells, which were transduced with shRNAs against OTUD6B, LIN28B or respective controls. Strikingly, we found a significant downregulation of prominent MYC targets (Mootha *et al*, 2003; Subramanian *et al*, 2005) in OTUD6B and LIN28B depleted MM cells, suggesting

a direct impact of the OTUD6B-LIN28B axis on MYC expression and activity (Fig 5A and B, and Appendix Fig S4A). Other known LIN28B targets such as E2F and mTORC1 hallmark genes were also affected by OTUD6B inactivation, further validating LIN28B as a relevant substrate of OTUD6B (Appendix Fig S4B and C; Zhu *et al*, 2011; Shyh-Chang & Daley, 2013).

MYC activation is a key event in the progression of the premalignant MGUS state to symptomatic MM and high MYC protein

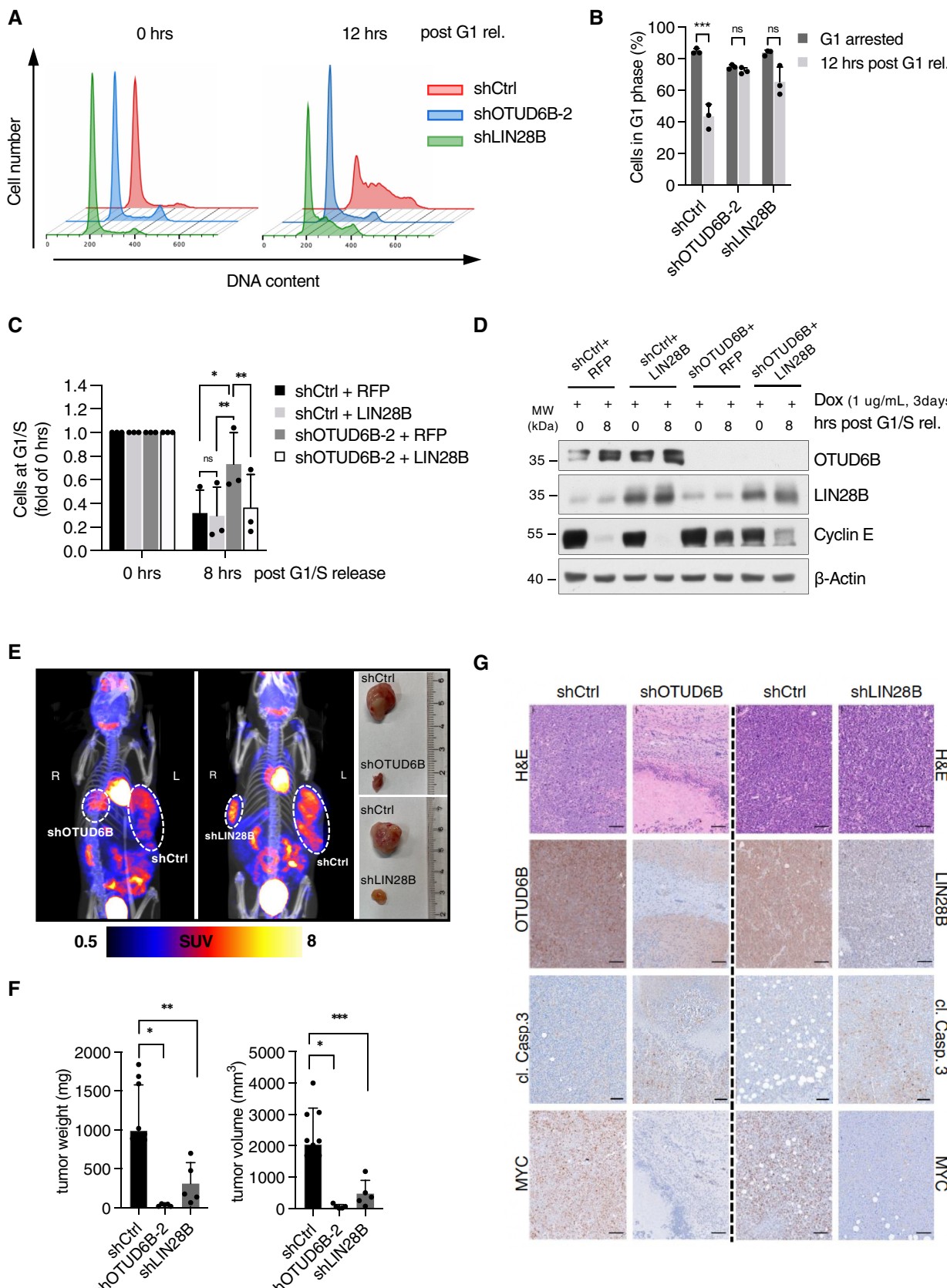

**Figure 4.**

◄

**Figure 4. The LIN28B-OTUD6B axis is a vulnerability in MM that drives cell cycle progression *in vitro* and *in vivo*.**

A, B Cell cycle analysis of MM1.S cells expressing the indicated shRNAs before and after G1-release. Cells were synchronized in late G1-phase using palbociclib and subsequently released for 12 hrs. DNA was stained with PI and analyzed by flow cytometry. (A) Exemplary FACS-plots at 0 and 12 h post G1 release. (B) Quantification of three biologically independent experiments showing the percentage of cells that remained in G1-phase after release ($n = 3$).

C Rescue experiment using MM1.S cells expressing the indicated shRNAs and doxycycline-inducible constructs of either RFP (EV) or LIN28B in which cell cycle analysis was performed before and after G1/S-release. MM1.S cells stably expressing doxycycline inducible RFP or LIN28B were transduced with either shCtrl or shOTUD6B. Cells were then synchronized at G1/S and transgene expression induced by doxycycline. DNA-content was analyzed before and 8 h after G1/S-release by PI staining and flow cytometry. Values are normalized to cells at G1/S before release. ($n = 3$ independent experiments).

D Immunoblot analysis of WCE obtained from a representative experiment depicted in (C).

E [18]F-FDG-PET/CT analysis of NOD.CB17/AlhnRj-*Prkdc^scid^*/Rj mice subcutaneously transplanted with human RPMI8226 cells expressing the indicated shRNAs (left two panels) and exemplary images of explanted tumors (right panel). At week 3–4 after transplantation, 5–10 MBq [18]F-FDG was administered intravenously and a 15-min static image was acquired 45 min after injection for each mouse.

F Metric tumor weight (left) and tumor volume (right) of the tumors derived from mice described in (E) after necropsy ($n = 8$ tumors shCtrl, $n = 4$ tumors shOTUD6B, $n = 4$ tumors shLIN28B).

G Immunohistopathology of representative tumors derived from sacrificed mice in (E) to visualize morphology (H&E), and expression of the indicated proteins. Scale bars, 100 µm.

Data information: In (B, C, and F), data represent mean ± s.d. *$P < 0.05$; **$P < 0.01$; ***$P < 0.001$ by paired Student's *t*-test (B, C, F) corrected for multiple testing (C). Source data are available online for this figure.

expression determines adverse outcome in MM (Chng *et al*, 2011; Dechow *et al*, 2014; Jovanovic *et al*, 2018). Different mechanisms of MYC activation in MM have been proposed, including genetic alterations (translocations and gains) and (post)transcriptional means. The contribution of these individual mechanisms to the net MYC protein abundance and transcriptional activity has remained elusive. We therefore evaluated the impact of OTUD6B on MYC expression in more detail. Depletion of OTUD6B significantly reduced MYC mRNA and protein levels in various MM cell lines to a similar extent irrespective of the nature of MYC aberrations present in the individual lines (Fig 5C; Dib *et al*, 2008; Quentmeier *et al*, 2019). Of note, silencing of both OTUD6B and LIN28B also led to a loss of MYC expression in our *in vivo* xenograft experiments (Fig 4G). Simultaneous depletion of OTUD6B and LIN28B did not further reduce MYC expression compared with the single knockdowns, supporting the notion that LIN28B functions downstream of OTUD6B (Fig 5D and E). Likewise, the reduction in cell proliferation was not further enhanced by a dual knockdown of LIN28B and OTUD6B (Fig 5F), while doxycycline induced re-expression of LIN28B partially restored MYC mRNA-levels in OTUD6B depleted MM cells (Appendix Fig S5D).

These data suggest that OTUD6B augments MYC protein expression via LIN28B and serves as a regulator of MYC by linking the UPS with post-transcriptional control.

### OTUD6B associates with MYC activity, MGUS to MM transition and poor outcome in MM patients

Finally, we investigated OTUD6B in MM patients and analyzed the mRNA levels of OTUD6B and MYC in primary CD138[+] cells from 89 patients with newly diagnosed symptomatic MM and found significant positive correlation, suggesting that OTUD6B regulates MYC expression in these patients (Fig 6A). In addition, analyses of two gene expression profiles of MM patient cohorts [GSE24080 ($n = 554$) and CoMMpass ($n = 786$)] revealed a significant enrichment of MYC target genes in patients with high OTUD6B expression (Fig 6B). We therefore asked whether OTUD6B has prognostic relevance in MM patients and found high OTUD6B expression to be associated with a significantly adverse overall survival (Fig 6C and D). In line with the role of MYC activation in MM progression, we

discovered a significant increase in OTUD6B expression along the transition from MGUS to MM (Fig 6E; Zhan *et al*, 2007). Of interest, OTUD6B expression did not correlate with any of the major clinically relevant prognostic cytogenetic abnormalities in our MM patient cohort (Fig EV5). We also found high OTUD6B expression to correlate with significantly reduced progression-free survival in patients treated with the proteasome inhibitor bortezomib, which is approved for all treatment lines of MM (Fig 6F; Mulligan *et al*, 2007; Kumar *et al*, 2017; Manasanch & Orlowski, 2017). In addition, knockout of OTUD6B lead to a significantly enhancement of the anti-myeloma activity of bortezomib and carfilzomib when using sub-lethal doses of the drugs (Fig 6G and H). OTUD6B may therefore contribute to the resistance towards proteasome inhibitory therapies.

These data validate OTUD6B as a new oncogene, dependency, and prognostic factor in MM that determines MYC activity in MM patients and may contribute to the conversion of premalignant MGUS state to the proliferative MM state.

## Discussion

This study identifies the deubiquitylase OTUD6B as a new vulnerability and oncogene in MM that exerts its specific activity towards the RNA binding protein LIN28B and eventually serves as a regulator of MYC activity to drive cell cycle progression. These results address the long-standing questions in how far DUBs can serve as actionable therapeutic targets in cancer, particularly in patients with MM and how the UPS can contribute to the conversion of MGUS to MM.

Targeting OTUD6B provides a strategy to activate let-7 microRNAs via LIN28B destabilization to repress their targets, of which we show that MYC is the prominent determinant in MM. While MYC has remained largely undruggable, OTUD6B inhibition would be a highly effective strategy to inactivate MYC (Dang *et al*, 2017). From a cell biological point of view, the OTUD6B-LIN28B nexus links proteolytic ubiquitylation to mRNA biogenesis and eventually directs MYC activity to the G1/S cell cycle transition, a previously unappreciated crosstalk (Hildebrandt *et al*, 2017). As such, OTUD6B acts as a superordinate regulator of MYC whose inhibition can

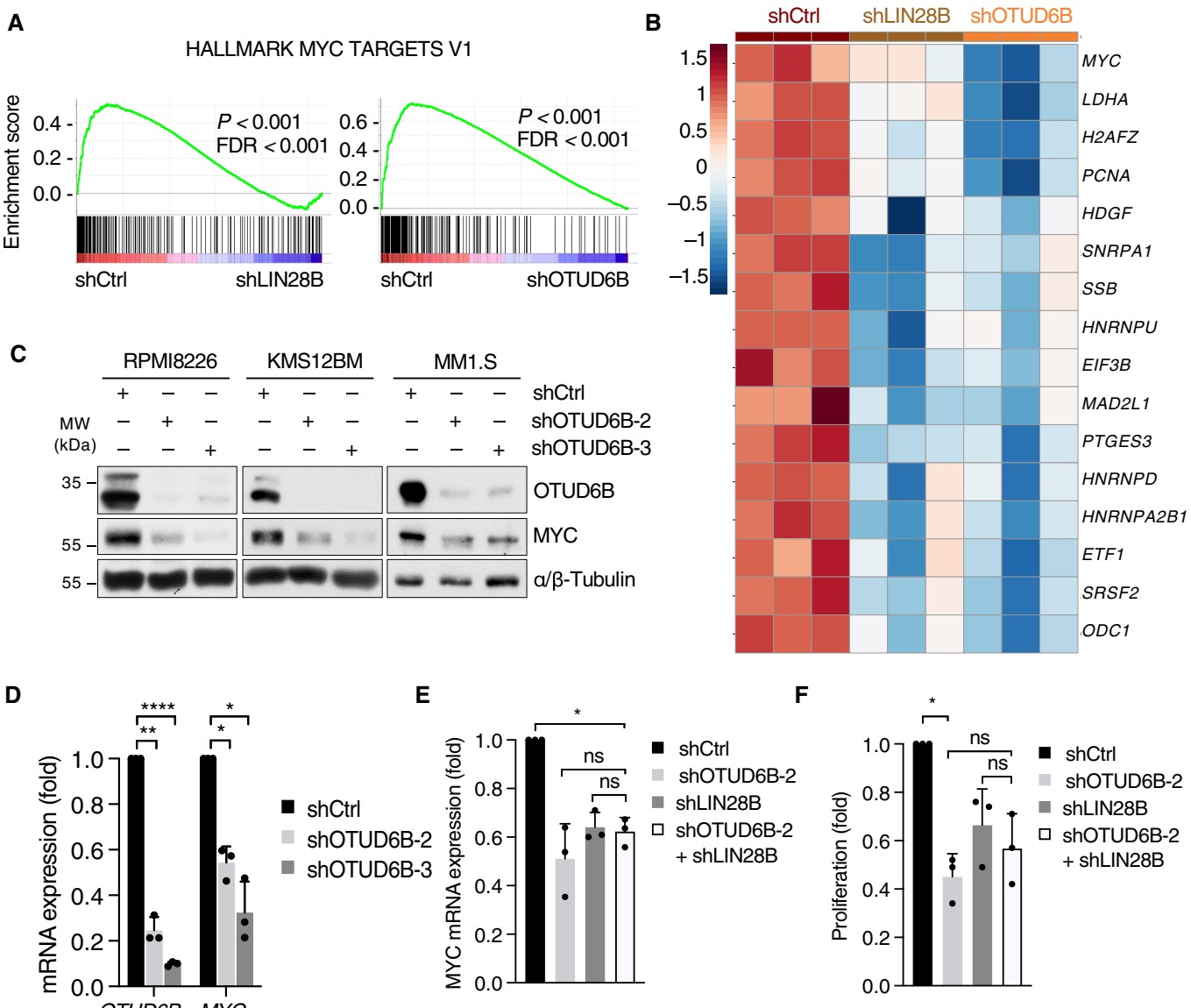

**Figure 5. OTUD6B activates MYC via LIN28B.**

A  GSEAs showing an enrichment of MYC V1 target genes in RPMI8226 MM cells depleted of OTUD6B and LIN28B when compared with control cells. RPMI8226 cells expressing the indicated shRNAs from three independent experiments were subjected to RNA-seq analysis followed by GSEA.

B  Heat map of MYC-regulated genes which were commonly downregulated ($P < 0.05$) in OTUD6B and LIN28B knockdown cells compared with control cells in the RNA-Seq experiment shown in (A) ($n = 3$).

C  Immunoblot analysis of RPMI8226, KMS12BM and MM1.S cells expressing the indicated shRNAs. Depicted are examples of three independent experiments per cell line. Specification of MYC rearrangements: RPMI8226 (der(17)t(?8;17)(q21.2;q25)), KMS12BM (der(1;8)(q10;q10)x2), MM1.S (der3t(3;8)).

D  Real-time qPCR of RPMI8226 cells in which OTUD6B expression was silenced by the indicated shRNAs.

E, F  Real-time qPCR (E) and proliferation (F) analyses of RPMI8226 cells infected with the indicated shRNA constructs.

Data information: In (D–F), values are normalized to shCtrl ($n = 3$ independent experiments with three technical replicates each). Data are mean ± s.d. *$P < 0.05$; **$P < 0.01$; ****$P < 0.0001$; by Student's *t*-test corrected for multiple testing.

Source data are available online for this figure.

abrogate MYC activity regardless of the nature of individual genetic MYC aberrations. While we largely demonstrate this effect in MM cells, we also demonstrate activity of the OTUD6B directed nexus in epithelial lung cancer cells, implying a more general functional role of this mechanism beyond MM.

DUBs of the OTU family carry catalytic cysteine sites that determine activity and are becoming increasingly amenable for inhibition (Mevissen & Komander, 2017; Harrigan *et al*, 2018). Targeted inhibition of OTUD6B would therefore not only seem technically feasible, but would provide an approach with a therapeutic window

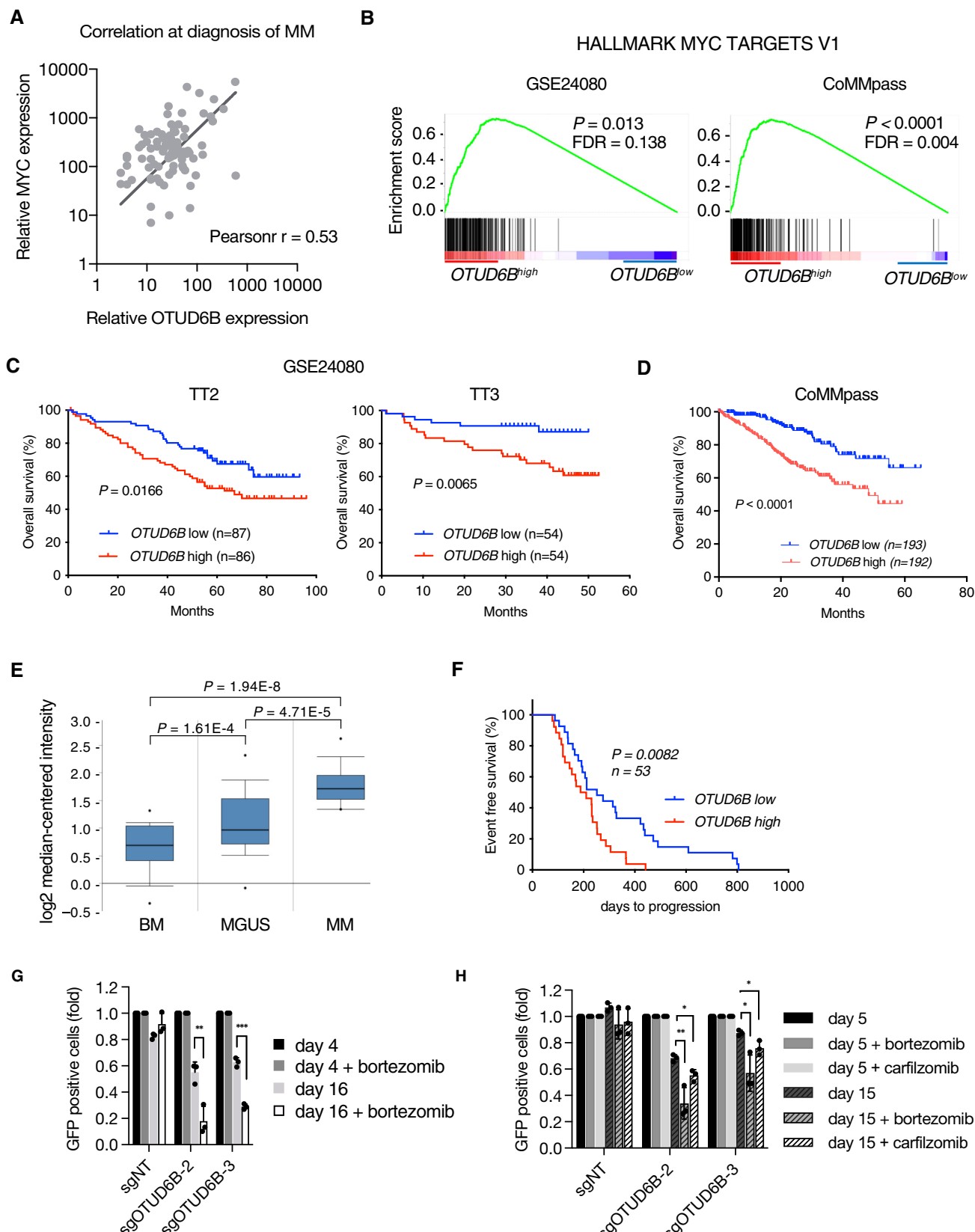

**Figure 6.**

**Figure 6. OTUD6B associates with MYC activity, MGUS to MM transition and poor outcome in MM patients.**

A    Correlation between *MYC* and *OTUD6B* expression in MM patient samples at diagnosis. mRNA expression in primary CD138[+] MM cells was quantified by real-time qPCR (*n* = 89 patients). Data are fit by linear regression (black line); Pearson *r*, Pearson correlation coefficient; *P* < 0.0001; by linear regression and Pearson correlation.

B    GSEAs showing an enrichment of MYC target genes in MM patients with high *OTUD6B* expression in two independent data sets (Left: GSE24080; Right CoMMpass-cohort − dbGaP accession: phs000748.v4.p3).

C, D  Kaplan–Meier survival curves of MM patients with high or low *OTUD6B* expression. Patients from the upper and lower quartiles of the two cohorts comprising the GSE24080 dataset (*n* = 176 for TT2 and *n* = 108 for TT3) (C) and the CoMMpass cohort (*n* = 385) (D) described in (B) were taken for analysis. *P* values calculated by Log-rank test.

E    Data from Zhan *et al* (2007) reanalyzed to show expression levels of *OTUD6B* in normal bone marrow (BM; *n* = 22), monoclonal gammopathy of undetermined significance (MGUS; *n* = 44) and multiple myeloma (MM; *n* = 12). Box-and-whisker plots show the upper and lower quartiles (25–75%) with a line at the median, whiskers extend from the 10[th] to the *n* percentile, and dots correspond to minimal and maximal values.

F    Kaplan–Meier curves for EFS of bortezomib treated MM patients with high or low *OTUD6B* expression. Patients classified as bortezomib responders (R) in the GSE9782 dataset were grouped according to their *OTUD6B* expression (highest and lowest 30%; *n* = 53) and analyzed for event free survival. *P* values calculated by Log-rank test.

G, H  Proliferation analysis of (G) LP-1-Cas9 and (H) MM1.S-Cas9 cells infected with indicated sgRNAs cultured without or with a sub-lethal dose of bortezomib (LP1: 7 nM; MM1.S: 12.5 nM) or carfilzomib (for MM1.S only. Sub-lethal dose of 5 nM). The ratio of sgRNA expressing (GFP[+]) cells to uninfected cells was measured by flow cytometry at the indicated time points after infection. Results are normalized to GFP[+] cells at day 4 (*n* = 3 independent experiments). Values represent mean ± s.d.

Data information: *P < 0.05; **P < 0.01; ****P < 0.0001; by Student's *t*-test, (H) corrected for multiple testing.
Source data are available online for this figure.

towards the Achilles heel of MM. The rationale for a potentially favorable toxicity profile of OTUD6B directed therapies stems from LIN28B, the substrate of OTUD6B, which is typically only expressed in embryonic cells or in certain malignancies such as MM (Shyh-Chang & Daley, 2013; Balzeau *et al*, 2017; Manier *et al*, 2017). Similarly, OTUD6B has been implicated in embryogenesis and developmental processes (Santiago-Sim *et al*, 2017; Straniero *et al*, 2018). Therefore, OTUD6B inhibition would be predicted to have limited effects on somatic cells and thus limited toxicity but teratotoxic effects of OTUD6B inhibition should be considered. Currently we have no evidence for the presence of other OTUD6B deubiquitylation substrates. However, LIN28B reconstitution in MM cells did not completely reverse the OTUD6B depletion cell cycle phenotype and we thus cannot fully rule out a contribution of other substrates that may associate with so far unanticipated side effects of OTUD6B inhibition.

The importance of such toxicity considerations is exemplified by USP14 and UCHL5, two proteasome-associated DUBs, whose inhibition demonstrated promising preclinical evidence to overcome bortezomib resistance in MM (Tian *et al*, 2014). However, a subsequent early phase clinical trial of VLX1570, a first in class DUB inhibitor targeting USP14 was terminated due to substantial pulmonary toxicity, likely attributable to the essentiality of USP14 to the overall proteasome function and thus lacking a sufficient therapeutic index (Rowinsky *et al*, 2020).

An OTUD6B targeting approach would also seem attractive in combination with proteasome inhibitors, given the outlined adverse effect of high OTUD6B expression on bortezomib-based therapies and the demonstrated synergy between bortezomib treatment and OTUD6B inactivation. In principle, proteasomal inhibition would be expected to exert stabilizing effects on LIN28B thereby potentially counteracting effects of OTUD6B depletion. We argue that in constellations of high OTUD6B expression in MM patients, inhibition of the proteasome likely further augments LIN28B expression to alleviate the efficacy of bortezomib thus possibly explaining the adverse outcome of these patients. By contrast, we speculate that OTUD6B inhibition outweighs the net effect of bortezomib treatment on LIN28B under clinically achievable bortezomib levels and that this

effect adds to the general proteotoxic stress induced by bortezomib in order to explain the observed synergy of OTUD6B inhibition and proteasomal inhibition.

Unbiased screening identified OTUD6B as a vulnerability in MM. In the pathophysiological context of this disease, this finding is intriguing given its function as a DUB that promotes S-phase entry. MM derives from the pre-malignant MGUS state in which cells typically contain most of the mutations found in active MM but rest in a dormant G1 state (Chng *et al*, 2011; van Nieuwenhuijzen *et al*, 2018). We find OTUD6B expression to increase as the disease undertakes the transition to active MM and we propose that the OTUD6B-LIN28B-MYC axis contributes to this malignant event to enable MM outgrowth. OTUD6B expression may thus guide future early intervention strategies upon prospective evaluation in clinical trials.

It is intriguing to speculate, that MM evolution resulted in a mechanism in which genetic re-expression of LIN28B and the rise of OTUD6B expression in MM coincide in order to enhance and time LIN28B activity and abundance to the G1/S transition to promote the conversion to active MM. This constellation then appears to have substantial impact on the maintenance and outcome of the disease. Indeed, we show that OTUD6B inactivation nearly completely abrogates MM growth *in vivo* via cell cycle arrest at the G1/S checkpoint that associates with a loss of MYC expression and that high OTUD6B expression associates with an adverse outcome of the disease.

The development of OTUD6B inhibitors thus seems warranted and we here present the biological and clinical framework and rationale from which to further approach this effort to improve the outcome of MM and potentially other tumors with an activated OTUD6B-LIN28B axis.

# Materials and Methods

### Cell culture and drug treatments

HEK293T, A549 and MM1.S were purchased from the American Type Culture Collection (ATCC). RPMI8226, H929, JJN3, KMS12BM,

LP-1, L363 and U266 were obtained from the German Collection of Microorganisms and Cell Cultures (DSMZ) and cultured according to the supplier's recommendations. All cell culture media were supplemented with 1% penicillin/streptomycin (Gibco). Indicated drugs were used at following concentrations: biotin 50 µM (Sigma), bromo deoxyuridine (BrdU) 10 µM (Sigma), cycloheximide 100–200 µg/ml (Sigma), MG132 10 µM (Tocris), nocodazole 500 ng/ml (Sigma), palbociclib 1 µM (Sigma), thymidine 2 mM (Sigma), doxy-cycline 1 µg/ml (Sigma), bortezomib 7–12.5 nM (Janssen-Cilag), carfilzomib 5 nM (SelleckChem). Cells were synchronized at G1/S or G2/M as described previously (Dietachmayr et al, 2020) by a double thymidine or a sequential thymidine and nocodazole block respectively. Palbociclib was used to synchronize cells at the G1 restriction point. If indicated, cells used for in vivo ubiquitylation or cycloheximide assays were treated with MG132 for 3 or 5 h, respectively, before harvesting. Doxycycline was used to induce LIN28B expression from the tet-on promotor present in pTRIPZ for the indicated duration at 1 µg/ml final concentration. Proliferation and cell viability were determined using the trypan-blue-exclusion method.

### Primary multiple myeloma cells

Patient derived MM cells were obtained in accordance with the ethical standards of the institutional and national research committee and with the 1964 Helsinki Declaration and its later amendments or comparable ethical standards. The investigation was approved by the Local Ethics Committee of our University Hospital (ethical approval # 438/19S). Written informed consent was obtained from each patient.

### Plasmids, shRNAs, and sgRNAs

Catalytic inactive mutant of OTUD6B (OTUD6B$^{C158A}$) was generated by site directed mutagenesis of OTUD6B cDNA. cDNAs of OTUD6B isoform 1 (IF1), OTUD6B isoform 2 (IF2), OTUD6B$^{C158}$, OTUD6A, OTUD2, OTUB1, LIN28B and LIN28B fragments (amino acid (AA) residues 29–250, 1–102, 1–166, 103–250 and 167–250) were cloned into pcDNA3.1 (Life Technologies) with or without sequences encoding an N-terminal HA or FLAG tag. LIN28B and OTUD6B were cloned into the pHIV-EGFP plasmid (Bryan Welm & Zena Werb; Addgene #21373) and pTRIPZ (Thermo Fisher Scientific; Kind gift of M. Reichert) and cDNA sequence coding for OTUD6B was cloned into pGEX-4T2 and pHIV-puro (modified from Addgene #21373). For BioID2 experiments, OTUD6B IF1 and IF2 were cloned into myc-BioID2-MCS (Kyle Roux; Addgene #74223). All cDNAs were sequenced. pRK5-HA-Ubiquitin-WT and pRK5-HA-Ubiquitin-K48 were obtained from Addgene (Ted Dawson; Addgene #17608). For shRNA-mediated silencing, shRNAs were cloned into pLKO.1 TRC (David Root; Addgene #10878) or into a pLKO.1 TRC, in which the puromycin-resistance cassette was replaced by DsRed-Express2. The following shRNA target sequences were used: shOTUD6B-1 (5′-CAGCTAGACAGTTAGAAATTA-3′), shOTUD6B-2 (5′-TGGCTTAGGAGAACATTATAA-3′), shOTUD6B-3 (5′-GATTTGTCTTACCAGATATTT-3′), shLIN28B (5′-GCAGGCATAATAAGCAAGTTA-3′) and shCtrl (5′-CCTAAGGTTAAGTCGCCCTCG-3′). To induce sgRNA-mediated knockout of OTUD6B in Cas9 expressing cells, the following sgRNA sequences were cloned into lentiGuide-GFP: sgOTUD6B-1 (5′-CTCAGCGGTCTGACTTCTCA-3′), sgOTUD6B-2 (5′-TACATACAGTGG

CCATCAGA-3′), sgOTUD6B-3 (5′-GAAGCAACTCACCGAAGATG-3′). The following non-targeting sequence was used: sgNT (5′-ACGGAGGCTAAGCGTCGCAA-3′).

### Transient transfections and lentiviral transductions

Transient transfections were done as described previously (Basser-mann et al, 2008), with Lipofectamine 2000 or Lipofectamine RNAiMAX (both Invitrogen). OTUD6B siRNA (#J-008553-05), LIN28B siRNA (#L-028584-01) and siCtrl (control) siRNA directed against luciferase (5′-CGTACGCGGAATACTTCGA-3′) were obtained from Dharmacon. Lentiviral transductions of cells were done as described previously (Heider et al, 2021). Briefly, cells were plated in virus-containing media supplemented with 8 µg/ml polybrene and spun at 216 g for 30 min. For stable expression of pHIV-Puro- or pTRIPZ-constructs, cell lines were selected with 0.3–1 µg/ml puromycin. For stable expression of Cas9, lentiCas9-Blast trans-duced cells were selected with 3–10 µg/ml blasticidin. Unchanged cell proliferation of Cas9 positive cell lines compared with wild-type cells was confirmed by cell counting assays.

### Construction of DUB library and DUB CRISPR screen

The DUB CRISPR library was designed to target all 98 DUBs and five genes essential for cell survival with three sgRNAs per gene and contained 12 non-targeting sgRNAs. Sequences were taken from the pooled human CRISPR-knockout library (GeCKOv2; Sanjana et al, 2014) and cloned into the lentiGuide-eGFP vector of the GeCKO-system. Lentiviral particles of the pooled library were produced and transduced into Cas9-expressing MM1.S cells at a ~2,000× sgRNA coverage. Cells were expanded for 2 days and 2 × 10$^6$ GFP$^+$ cells sorted using a FACSAria (BD Biosciences). Half was harvested as T0 sample, while the other was cultured for 14 days before sample collection (T14). Genomic DNA was extracted, sgRNA cassettes amplified by a two-step PCR approach, adding adapters and sample barcodes for deep-sequencing (Illumina). Products were quantified by qPCR using the KAPA Library Quantification Kit (Kapa Biosystems). Deep-sequencing of samples was performed on a MiSeq Illumina machine using the MiSeq Reagent Kit v2 (Illumina). Reads were aligned to the sgRNA sequence library with bwa 0.7.12 (preprint: Li, 2013). Read counts were then determined for each guide RNA with samtools v1.12. After normalization of sgRNA-reads to the total number of reads, enrichments and dropouts were calculated between T0 and T14 samples.

### Flow cytometry

The percentage of GFP/dsRed$^+$ cells within a population was determined by flow cytometry on a FACSCalibur or Accuri C6plus (Becton Dickinson). For measurement of DNA content, cells were fixed in ice-cold 70% ethanol at −20°C and incubated in PI/RNase staining buffer (BD Pharmingen) according to the manufacturer's description. For BrdU/PI cell cycle analyses, cells were treated with BrdU for 40 min, washed twice with PBS and fixed in ice-cold 70% ethanol at −20°C. FITC Mouse Anti-BrdU (clone B44; BD Biosciences) was used to stain cells in S-phase according to the manufacturer's description. DNA was stained with 1 µg/ml propidium

iodide (PI) and samples were analyzed on a FACSCalibur or Accuri C6plus (Becton Dickinson). The resulting data were analyzed using FlowJo (TreeStar Inc.).

## Antibodies

The following antibodies were used for immunoblotting: α/β-tubulin (1:1,000, rabbit, Cell Signaling #2148), β-actin (1:5,000, mouse, clone AC-15, Sigma #A-1978), Caspase 3 (1:1,000, rabbit, clone 8G10, Cell Signaling #9665), Cleaved Caspase 3 (1:500, rabbit, clone 5A1E, Cell Signaling #9664), CUL1 (1:500, mouse, clone 2H4C9, Sigma #32-2400), Cyclin A (1:1,000, mouse, clone H-432, Santa Cruz #sc-751), Cyclin B1 (1:1,000, rabbit, Cell Signaling #4138), Cyclin E (1:1,000, mouse, clone HE12, Santa Cruz #sc-247), FLAG (1:1,000, rabbit, Sigma #F7425), HA (1:1,000, rabbit, Cell Signaling #3724), LIN28B (1:1,000–1:10,000, rabbit, Cell Signaling #4196), MYC (1:1,000, mouse, clone 9E10, Santa Cruz #sc-40), OTUD6B (1:1,000, rabbit, Abcam #ab127714), p27 (1:500, mouse, clone G173-524, BD Biosciences #554069), p53 (1:1,000, rabbit, Cell Signaling #9282), p-p53 (Ser15) (1:1,000, rabbit, Cell Signaling #9284), p-Histone H3 (Serine 10, 1:1,000, rabbit, Cell Signaling #9701), p-GSK-3β (Serine 9, 1:1,000, rabbit, Cell Signaling #9322), PLK1 (1:500, mouse, clone PL6/PL2, Thermo Fisher #33-1700), SKP2 (1:1,000, rabbit, Zymed #51-1900), Ubiquitin K48 (1:1,000, rabbit, clone Apu2, Millipore #05-1307). Secondary antibodies (anti–rabbit IgG and anti–mouse IgG, 1:5,000) and protein A coupled to horseradish peroxidase were purchased from GE Healthcare.

## Biochemical methods

Whole cell extract (WCE) preparation, immunoprecipitation, and immunoblotting have been described previously (Bassermann et al, 2008; Fernandez-Saiz et al, 2013; Baumann et al, 2014; Eichner et al, 2016). Briefly, cell pellets were lysed in lysis buffer (150 mM NaCl, 50 mM Tris-HCl pH 7.5, 5 mM MgCl$_2$, 1 mM EDTA, 0.1% NP-40, 5% glycerol and protease inhibitors) for 20 min on ice and clarified at 20,800 × g and 4°C for 20 min. For immunoprecipitations (IP), cell lysates were precleared with Protein G agarose (Millipore) for 30 min at 4°C and then incubated with FLAG-M2 agarose beads (Sigma) or HA agarose beads (Sigma) at 4°C for 1.5–2 h. Beads were washed four times with lysis buffer and boiled in 2× laemmli buffer for 5 min. *In vivo* ubiquitylation experiments were performed as previously described (Fernandez-Saiz et al, 2013; Baumann et al, 2014). Briefly, cells were lysed in buffer containing 250 mM NaCl, 50 mM Tris/HCl pH 7.5, 1 mM EDTA, 0.1% Triton X-100, 50 mM NaF and protease inhibitors, incubated for 10 min on ice and centrifuged at 20,800 × g and 4°C for 10 min. Samples were denatured by the addition of 1% SDS and 5 mM EDTA and subsequent boiling at 95°C for 5 min. After quenching with 1% Triton-X-100, samples were subjected to FLAG-IP. Induction and purification of GST-tagged proteins from *Escherichia coli* BL21 (Agilent, #200131) were performed as described previously (Heider et al, 2021). For GST pulldown experiments, purified bead-bound GST fusion-proteins or GST alone were incubated with clarified mammalian cell lysate for 1h at 4°C. Beads were then washed 4 times with lysis buffer and boiled in 2× laemmli buffer

for 5 min. Densitometry analysis was done using Licor Image Studio Lite 5.2.

## Mass spectrometric analyses

Mass spectrometric analyses to identify substrates of OTUD6B were performed both affinity-based (FLAG-purification) and proximity-based (BioID2). For FLAG-OTUD6B purification, 2 × 10$^9$ HEK293T cells were transfected with FLAG-EV or FLAG-OTUD6B. Lysis and immunoprecipitation were performed as described above with the following modifications. After washing the beads four times with lysis buffer and once with TBS, bound proteins were eluted with 1 mg/ml 3xFLAG peptide (Sigma) in TBS for 10 min at room temperature. Finally, proteins were precipitated overnight with 10% TCA at 4°C, washed with acetone and subsequently dried. Samples were reduced by DTT and alkylated by chloroacetamide. Peptides were generated by in-gel trypsin digestion and dried down. For BioID2 purification of OTUD6B IF1 and IF2, 10 × 15 cm dishes of HEK293T cells were left untreated or transfected with myc-BioID2-EV, myc-BioID2-OTUD6B-IF1 or -IF2, respectively. After 24 h, cells were treated with 50 μM biotin for 16 h. Cells were lysed in modified RIPA lysis buffer (50 mM Tris-HCl pH 7.5, 150 mM NaCl, 1 mM EDTA, 1 mM EGTA, 1% Triton X-100, 0.1% SDS, 0.5% sodium deoxycholate, protease inhibitors), passed three times through a 22G and once through a 26G syringe needle and incubated for 1 h on ice. Lysates were clarified and biotinylated proteins purified with strep-tactin superflow resin (IBA) for 3 h at 4°C. Beads were washed two times with lysis buffer and once with 50 mM ammonium bicarbonate buffer at pH 8.5. Proteins bound to beads were reduced by DTT and alkylated by chloroacetamide. On-bead trypsin digestion was performed overnight at 37°C and peptides were dried down.

   Tryptic peptides were reconstituted in 0.1% formic acid (FA) and analyzed by nanoLC-MS/MS on an Eksigent NanoLC-Ultra 1D+ system coupled to a Q Exactive HF mass spectrometer (Thermo Fisher Scientific) using a 100 min (FLAG-IP) or 50 min (BioID) linear gradient from 4 to 32% LC solvent B (0.1% FA, 5% dimethyl sulfoxide (DMSO) in acetonitril) in LC solvent A (0.1% FA in 5% DMSO). For FLAG-IP samples, MS1 spectra were recorded in the Orbitrap from 360 to 1,300 m/z at a resolution of 60K (automatic gain control (AGC) target value of 3e6 charges, maximum injection time (maxIT) of 10 ms). After peptide fragmentation via higher energy collisional dissociation (normalized collision energy of 25%), MS2 spectra for peptide identification were recorded in the Orbitrap at 30K resolution via sequential isolation of up to 20 precursors (isolation window 1.7 m/z, AGC target value of 2e5, maxIT of 50 ms, dynamic exclusion of 35 s). BioID samples were measured as specified above with following modifications: MS1 maxIT was set to 50 ms. MS2 spectra were recorded at 15k resolution using an AGC target value of 1e5 and a dynamic exclusion of 20 s. Peptide and protein identification and quantification for BioIDs were performed using MaxQuant 1.5.3.30 by searching the MS2 spectra against the human reference proteome supplemented with common contaminants. FLAG-IP raw data were searched using MaxQuant v1.5.6.5 and the SwissProt database. The match-between-runs and label-free quantification algorithms were enabled. Protein intensities were computed as the sum of the area-under-the-curve of chromatographic elution profiles of peptides assigned to the proteins.

## DUB activity assays

For assessment of DUB activity toward HA-ubiquitin vinyl sulfone (HA-Ub-VS), cells were lysed in DUB activity buffer (50 mM Tris/HCl pH 7.4, 5 mM MgCl$_2$, 250 mM sucrose, 1 mM DTT and 2 mM ATP) for 20 min on ice and lysates subsequently clarified. Equal amounts of protein were mixed with 5 µM HA-Ub-VS (Boston Biochem #U-212) and subsequently incubated at 37°C for 45 min. The reaction was stopped and lysates were denatured by the addition of 1% SDS and subsequent boiling at 95°C for 5 min. After cooling to room temperature, samples were diluted with DUB activity buffer to a final volume of 500 µl. Active DUBs modified by HA-Ub-VS were immunoprecipitated by HA agarose. Beads were washed four times with washing buffer (250 mM NaCl, 50 mM Tris/HCl pH 7.5, 1 mM EDTA, 0.1% Triton X-100, 50 mM NaF).

For *in vitro* DUB assays, FLAG-tagged LIN28B was co-expressed with HA-tagged ubiquitin in HEK293T cells, purified as described for *in vivo* ubiquitylation experiments, eluted with 3xFLAG peptide (Sigma #F4799) at a final concentration of 1 mg/ml in DUB buffer (50 mM Tris/HCl pH 7.4, 100 mM NaCl, 2 mM DTT) and used as substrate. GST or GST-OTUD6B bound to beads were washed three times with DUB buffer, activated for 10 min at 23°C and subsequently incubated with eluted substrate for 1.5 h at 37°C. The reaction was stopped by the addition of laemmli buffer and boiling at 95°C for 5 min.

## Xenograft experiments and IHC of tumor samples

For xenograft experiments, human RPMI8226 cells were lentivirally transduced with either shRNA constructs specifically targeting OTUD6B, LIN28B or scrambled control shRNA. $0.6 \times 10^7$ cells were suspended in serum free medium, mixed with Matrigel h.c. (Corning) at a 1:1 ratio and injected subcutaneously into the opposite flanks of randomly selected female NOD.CB17/AlhnRj-*Prkdc*$^{scid}$/Rj mice 8–10 weeks of age (Charles River). For all tumor growth studies, a group size of at least four animals per condition was chosen. Animals were censored from analyses when sacrificed for nontumor reasons. Mice were housed under SPF conditions and animal experiments were conducted in accordance with the local ethical guidelines with permission from the District Government of Upper Bavaria (application no.: 55.2-2532.Vet_02-16-141). Tumors were allowed to reach a predefined humane maximal size. Three to four weeks post xenograft implantation, mice underwent imaging using a micro-PET and CT small animal scanner (Mediso) at the preclinical imaging facility of TranslaTUM. A dose of 5–10 MBq $^{18}$F-FDG was administered by tail vein injection and a 15-min static PET image was acquired 45 min after injection, followed by CT. Images were analyzed using the Mediso nanoScan software and displayed as maximum intensity projections (MIPs) displaying the standardized uptake value (SUV), with overlay of the CT image for anatomical reference. At the end of the experiment, mice were sacrificed, xenograft tumors were explanted and subsequently measured and weighed. Tumor tissue samples for immunohistochemistry (IHC) were fixed in 4% PFA. For IHC, sections were deparaffinized and rehydrated. Heat mediated antigen retrieval was conducted in preheated target retrieval solution (Agilent), Pro Taqs II Antigen-Enhancer (Quartett) or Target Unmasking Fluid (PanPath) according to the supplier's recommendations. Subsequently slides were incubated with anti-OTUD6B antibody (1:60, Atlas Antibodies, #HPA024046), anti-LIN28B antibody (1:100, Atlas Antibodies, #HPA036630), anti-cl. Caspase antibody (1:100, Cell Signaling #9664) and anti c-MYC antibody (1:100, Klon Y69, Abcam, #ab32072) for 1 h at room temperature. Signal detection was performed using ImmPRESS Anti-Rabbit IgG Polymer Kit (Vector; for c-MYC, cl. Caspase and OTUD6B) or MACH 3 Rabbit HRP Polymer Detection (Biocare Medical, for LIN28B) with DAB+ chromogen (Agilent) according to the suppliers' recommendations. Counterstaining was performed using Hematoxylin Gill's Formula (Vector).

## RNA-Seq analysis

Library preparation for bulk-sequencing of poly(A)-RNA was done as described previously (Parekh *et al*, 2016). Briefly, barcoded cDNA of each sample was generated with a Maxima RT polymerase (Thermo Fisher) using oligo-dT primer containing barcodes, unique molecular identifiers (UMIs) and an adaptor. Ends of the cDNAs were extended by a template switch oligo (TSO) and full-length cDNA was amplified with primers binding to the TSO-site and the adaptor. NEB UltraII FS kit was used to fragment cDNA. After end repair and A-tailing a TruSeq adapter was ligated and 3'-end-fragments were finally amplified using primers with Illumina P5 and P7 overhangs. In comparison with Parekh *et al* (2016), the P5 and P7 sites were exchanged to allow sequencing of the cDNA in read1 and barcodes and UMIs in read2 to achieve a better cluster recognition. The library was sequenced on a NextSeq 500 (Illumina) with 63 cycles for the cDNA in read1 and 16 cycles for the barcodes and UMIs in read2. Gencode gene annotations M25 and the human reference genome GRCh38 were derived from the Gencode homepage (EMBL-EBI). Drop-Seq tools v1.12 was used for mapping raw sequencing data to the reference genome (Macosko *et al*, 2015). The resulting UMI filtered count matrix was imported into R v4.0.5. Technical replicates were summarized by summing up the read-counts per gene and lowly expressed genes were subsequently filtered out. Prior differential expression analysis with DESeq2 v1.18.1 (Love *et al*, 2014), dispersion of the data was estimated with a parametric fit using an univariate model where treatment was specified as independent variable. The Wald test was used for determining differentially regulated genes between treatments and shrunken log2 fold changes were calculated afterwards. A gene was determined as differentially regulated if the absolute apeglm shrunken log2 fold change was at least 1 and the adjusted *P*-value was below 0.01. GSEA v4.0.3 was performed in the weighted pre-ranked mode where the apeglm shrunken foldchange was used as ranking metric (Subramanian *et al*, 2005). All genes tested for differential expression were used for GSEA analysis with gene sets from MsigDB v7.4 (Liberzon *et al*, 2011, 2015). A pathway was considered to be significantly associated with a treatment if the FDR value was below 0.05. Rlog transformation of the data was performed for visualization and further downstream analysis. Heatmaps show z-transformed expression data. Raw sequencing data is available from the European Nucleotide Archive under the accession number PRJEB45829.

## mRNA analysis

To analyze mRNA expression, RNA was extracted and reverse-transcribed to cDNA with the Superscript III Reverse Transcriptase

(Invitrogen). Real-time quantitative PCR analysis was performed on a LightCycler 480 (Roche) using the following primer sequences: OTUD6B (forward 5′-ATTGACCGAAGAGCTTGATGAGG-3′, reverse 5′-TTGGCTTGCAACTCCTTCTTCTC-3′), MYC (forward 5′-TCAAGAGGCGAACACACAAC-3′, reverse 5′-GGCCTTTTCATTGTTTTCCA-3′) and RPLP0 (forward 5′-GATTGGCTACCCAACTGTTG-3′, reverse 5′-CAGGGGCAGCAGCCACAAA-3′). mRNA levels of RPLP0 was used as a reference gene.

## Statistical analyses

Statistical evaluations of data sets were performed by log-rank (Mantel-Cox) test, paired or unpaired two tailed Student's *t*-test, Pearson's correlation or linear regression or one-way ANOVA, according to assumptions of the test using GraphPad Prism (GraphPad Prism 9) software. *P*-values were adjusted using the Bonferroni-Dunn method integrated in the Prism software for multiple testing. The error bars shown in the figures represent the mean ± s.d., unless specified otherwise. The *P* values presented in the figures and legends (when a statistically significant difference was found) are: $*P < 0.05$, $**P < 0.01$, $***P < 0.001$ and $****P < 0.0001$.

# Data availability

The CRISPR drop out screen and RNASeq raw data have been deposited on the ENA server (https://www.ebi.ac.uk/ena) with the following accession numbers: PRJEB46352 (sgRNA screen; https://www.ebi.ac.uk/ena/browser/view/PRJEB46352) and PRJEB45829 (RNASeq).

The mass spectrometry proteomics raw data have been deposited on the PRIDE server (https://www.ebi.ac.uk/pride/archive/) with the following accession number: PXD027480. Further primary data will be made publicly accessible upon publication or reviewer request.

**Expanded View** for this article is available online.

## Acknowledgements

We thank Markus Utzt for cell sorting, the Preclinical Imaging Core at TranslaTUM (PICTUM; especially Markus Mittelhäuser and Hannes Rolbieski) and the Center of Preclinical Research at Klinikum rechts der Isar der Technischen Universität München for their support with xenograft experiments and imaging. This work was supported by grants from the European Research Commission—project BCM-UPS, grant #682473 to FB, the Deutsche Forschungsgemeinschaft (DFG, German Research Foundation)—Project-ID 360372040-SFB 1335 to FB and UK, BA 2851/6-1 to FB and the Deutsche Krebshilfe 70114425 to UK. Open access funding enabled and organized by Projekt DEAL.

## Author contributions

**Carmen Paulmann:** Conceptualization; data curation; formal analysis; methodology; writing – original draft. **Ria Spallek:** Conceptualization; data curation; formal analysis; validation; visualization; methodology; writing – original draft; writing – review and editing. **Oleksandra Karpiuk:** Resources; methodology. **Michael Heider:** Data curation; visualization; methodology. **Isabell Schäffer:** Data curation; validation; methodology. **Jana Zecha:** Data curation; formal analysis; validation; methodology. **Susan Klaeger:** Data curation; formal analysis; validation; methodology. **Michaela Walzik:** Data curation. **Rupert Öllinger:** Software; formal analysis; methodology. **Thomas Engleitner:** Data curation; software; visualization; methodology. **Matthias Wirth:** Resources; data curation; visualization. **Ulrich Keller:** Conceptualization; data curation. **Jan Krönke:** Data curation; formal analysis; visualization; methodology. **Martina Rudelius:** Data curation; visualization; methodology. **Susanne Kossatz:** Data curation; visualization; methodology. **Roland Rad:** Conceptualization; data curation. **Bernhard Kuster:** Conceptualization; formal analysis; validation; visualization; methodology. **Florian Bassermann:** Conceptualization; formal analysis; supervision; funding acquisition; validation; investigation; visualization; methodology; writing – original draft; project administration; writing – review and editing.

In addition to the CRediT author contributions listed above, the contributions in detail are:
FB initiated this project. CP, RS and FB conceived and designed the research; CP and RS performed the experiments with crucial help from IS and MWa; OK designed and cloned the CRISPR library; JZ, SKl and BK performed mass spectrometry and bioinformatic workup; TE, RO and RR performed RNA-Seq and Illumina Miseq analyses; MWi performed GSEAs of MM data sets; JK analyzed mRNA expression of MM samples; MH and SKo conducted the xenograft experiments and imaging and MR provided IHC analysis of the respective tumor samples. UK provided important conceptual advice. CP, RS and FB wrote the manuscript; All authors discussed the results and commented on the manuscript.

## Disclosure and competing interests statement

F.B. and U.K. received honoraria and research funding from BMS/Celgene.

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
