## [Review Process File · The EMBO Journal]

The OTUD6B-LIN28B-MYC axis determines the proliferative state in multiple myeloma

Carmen Paulmann, Ria Spallek, Oleksandra Karpiuk, Michael Heider, Isabell Schäffer, Jana Zecha, Susan Klaeger, Michaela Walzik, Rupert Öllinger, Thomas Engleitner, Matthias Wirth, Ulrich Keller, Jan Krönke, Martina Rudelius, Susanne Kossatz, Roland Rad, Bernhard Küster, and Florian Bassermann

DOI: [10.15252/emj.2022110871](https://doi.org/10.15252/emj.2022110871)

Corresponding author: Florian Bassermann (florian.bassermann@tum.de)

Review Timeline:

Submission Date:	7th Feb 22
Editorial Decision:	7th Mar 22
Revision Received:	21st Jun 22
Editorial Decision:	20th Jul 22
Revision Received:	27th Jul 22
Accepted:	1st Aug 22

Editor: Kelly Anderson

Transaction Report:

Dear Prof. Bassermann,

Thank you for submitting your manuscript for consideration by the EMBO Journal. It has now been seen by three referees whose comments are shown below.

Given the referees' positive recommendations, I would like to invite you to submit a revised version of the manuscript, addressing the comments of all three reviewers. I should add that it is EMBO Journal policy to allow only a single round of revision, and acceptance of your manuscript will therefore depend on the completeness of your responses in this revised version. It would be good to discuss your plan for addressing the concerns, which I am happy to do by email or zoom in the next couple weeks.

I have attached a guide for revisions for your convenience.

Thank you for the opportunity to consider your work for publication. I look forward to your revision.

Yours sincerely,

Kelly M Anderson, PhD
Editor
The EMBO Journal
k.anderson@embojournal.org

We realize that it is difficult to revise to a specific deadline. In the interest of protecting the conceptual advance provided by the work, we recommend a revision within 3 months (5th Jun 2022). Please discuss the revision progress ahead of this time with the editor if you require more time to complete the revisions. Use the link below to submit your revision:

Link Not Available

Referee #1:

In this work, Paulmann and colleagues screened the deubiquitylases dependencies in MM. Using MM cell lines, they identified OTUD6B as an oncogene driving G1/S transition. They identified LIN28B as a substrate of OTUD6B. Their inhibition is associated with MYC depletion and proliferation inhibition. The biological effects were validated in vivo using xenograft model. Using publicly available GEP data, the authors identified that high OTUD6B expression is associated with a poor outcome in MM and a link with resistance to proteasome inhibitors.

This is an interesting study with extensive work.
However, important comments should be addressed to strengthen the manuscript:

Major comments:

1/ The results presented in the Figure 1B demonstrating the effect on cell cycle have been obtained with different cell lines. However, these cell lines are associated with the same cytogenetic abnormality resulting in MAF deregulation. MAF translocations/deregulations represent the smaller molecular subgroup in MM with 8% of the patients. The authors should validate the biological results of OTUD6B depletion on cell lines from other molecular subgroups reflecting the molecular heterogeneity of the disease.

What are the effects of OTUD6B depletion on MM cell survival?

The authors should validate the reversion of the phenotype with OTUD6B supplementation.
It would be interesting to know if OTUD6B results or not in DNA damages and DDR.

2/ The authors identified a downregulation of MYC after OTUD6B or LIN28B depletion. Since the MYC-IRF4 axis plays a major role in MM biology, the authors should investigate the effect on IRF4 expression.

3/ The authors used publicly available cohorts and identified that high OTUD6B expression is associated with a poor outcome. However, the GSE2658 dataset is composed of two different cohorts TT2 and TT3 with differences in the treatment. The authors should investigate the prognostic value in the TT2 and TT3 cohorts separately. Furthermore, the other dataset used GSE24080 comprised the same cohorts TT2 and TT3 than GSE2658.

The authors should try to identify another independent cohort of patients. The CoMMPASS cohort including newly diagnosed MM patients characterized by RNA sequencing could be of particular interest. In the KM figures, the authors should include the number of patients in each group.

The authors should also investigate the link between OTUD6B expression and the different cytogenetic abnormalities in MM. Since several cytogenetic events are associated with a prognostic value in MM, it will be important.

4/ The MYC/IRF4 axis is targeted by IMiDs (immunomodulatory agents) in MM. According to the results presented by the authors, it will be of interest to investigate if OTUD6B depletion could potentiate IMiD MM cell cytotoxicity.

5/ The authors presented that OTUD6B depletion potentiates bortezomib toxicity using LP1 cell line. This is not clear why the authors used another cell line compared to the results presented in Figure 1. These results should be validated at least on other MM cell lines. Furthermore, since validation on primary MM cells seems not feasible without specific OTUD6B inhibitor, an in vivo validation using xenograft model would be important.

It would be interesting to investigate the response to carfilzomib proteasome inhibitor.

Referee #2:

In this study, Paulmann et al identify the OTU-family deubiquitinase OTUD6B as a regulator of LIN28B stability in multiple myeloma cells. Starting from a CRISPR/Cas9 screen of 98 human DUBs to identify candidate genes that promote proliferation of multiple myeloma cells, they find that OTUD6B-depleted cells exhibited a decrease in proliferation and an increase in cells stalled at the G1/S phase transition point. In a well-designed series of experiments, the authors show that OTUD6B-mediated removal of K48 polyUb rescues LIN28B from proteolytic degradation at G1/S phase, leading to derepression of its microRNA targets including MYC and consequent rapid S-phase entry and enhanced proliferation. Through the use of a cysteine-reactive ubiquitin probe, the authors demonstrate that OTUD6B binding increases at G1/S while protein levels remain constant, suggesting that DUB activity is cell-cycle regulated. They supplement in vitro data with a mouse model of tumor growth and with patient data correlating OTUD6B expression with disease progression and outcome.

Although the authors present a largely convincing set of experiments to link OTUD6B with cell cycle-dependent expression of LIN28B, there is no indication throughout as to how many times experiments including western blots supporting several key findings were performed and this should be rectified. In particular, the data suggesting cell cycle-dependent modulation of OTUD6B catalytic activity would benefit from quantitation of multiple replicates (Fig 1F). If reproducible, this would be a striking observation & would merit further investigation.

Comments:

Fig 1A: Of the DUB family genes investigated in the CRISPR screen, OTUD6B knockdown lead to only a comparatively minor survival disadvantage with respect to other targets. It would be interesting to know what these were. Additionally, this minor disadvantage seems to be at odds with the data in Fig 1B, in which OTUD6B knockdown reduced mean proliferation by up to 90% in MM1.S cells.

There are several other DUBs showing a similar drop in representation, with 6 exhibiting a much larger reduction, with two of these showing a similar magnitude to some of the essential gene controls. From this panel alone, OTUD6B doesn't appear to be the best candidate to take forward from the screen. An explanation as to why OTUD6B was selected over these would be useful.

Fig 2B: It is not clear how proliferation was assessed. How many replicates were performed?

Fig 2C: It is not necessary to show the K63-specific blot, given that there is no detectable signal to confirm the antibody worked. This panel can be removed.

Overexpression of catalytically-inactive OTUD6B also appears to cause a reduction in polyubiquitylated LIN28B, albeit less than WT. This should be discussed. Again, these data would be easier to interpret if some indication of reproducibility were provided.

In the accompanying text, the authors state that OTUD6B (C158A) did not deubiquitylate LIN28B in the same experiment, however there does still appear to be a substantial reduction in the ubiquitin signal (both anti-HA and anti-K48) relative to those cells not expressing OTUD6B (compare lanes 3 and 5).

The conclusions would be stronger if the authors been able to demonstrate accumulation of endogenous LIN28B-pK48 upon 6B knockdown (& subsequent rescue). The LIN28B expressed in cells are all Flag-tagged, a tag which has lysine residues that may contribute to the strong ubiquitylation observed. Complementing the data shown with endogenous protein or identifying lysine residues in LIN28B that are ubiquitylated will address this concern.

Fig 2D: It is somewhat surprising that OTUD6B interacts with non-ubiquitylated LIN28B. Does it preferentially immunoprecipitate the polyUb form purified from cells in panel E?

Fig 3B-D: These would benefit from normalised quantitation of LIN28B expression from experimental replicates.

Fig 4B, C: It is not clear whether the individual data points presented represent technical replicates within a single experiment or pooled data from separate experiments. If the former, how many replicates were performed?

Fig 4C, D: The equivalent levels of LIN28B in RFP controls of both control and OTUD6B-silenced cells 8h post G1/S release is difficult to reconcile with the data at similar time points in Fig 3D, especially given the extension of G1/S phase in the knocked down cells. This should be addressed. Additionally, the maintenance of Cyclin E upon OTUD6B knockdown, even during overexpression of LIN28B, suggests the possible existence of additional OTUD6B substrates relevant for cell cycle progression. This data should be quantitated and discussed.

Fig 4G: The claimed increase in p27 expression in OTUD6B knockdown tumors is not convincing in the image shown. If anything, there appears to be more p27 staining in the control tumor.

Fig 6F: The preceding data identify OTUD6B-mediated rescue of LIN28B from proteolytic degradation as a promoter of enhanced proliferation in multiple myeloma cells. These effects are opposed by OTUD6B knockdown and consequent increased LIN28B degradation. Since proteasome inhibition would be expected to rescue LIN28B levels, it is somewhat surprising that the authors observed synergistic effects of bortezomib and OTUD6B knockdown upon suppression of proliferation. Any explanation or discussion?

Minor comments:

- In the figure 5A legend: GESA -> GSEA.

- The discussion feels lacking in the context of what is already known about OTUD6B biology, though admittedly there have been few high-quality studies. Mutations in OTUD6B, presumably resulting in a loss of function, are known to result in an intellectual disability disorder with dysmorphic features in humans, and corresponding KO is embryonically lethal in mice (Santiago-Sim et al, 2017 - PMID: 28343629). This is perhaps of relevance with LIN28B as a developmentally regulated protein, and worth considering in the context of OTUD6B inhibition?

Assuming these comments are properly addressed I would have no hesitation in recommending this manuscript for publication.

Reviewed by Yogesh Kulathu

Referee #3:

I have read with great interest the paper by Paulmann et al, entitled "The OTUD6B-LIN28B-MYC axis determines the proliferative state in multiple myeloma".

This is an exceptionally well crafted manuscript with scrupulous methodology, high degree of novelty and potential impact.

As a reviewer, I have hardly ever read a manuscript and found no major flaws to it. This work was a pleasure to read and I commend the authors for their thorough work.

I have several minor points that require either a compelling explanation or some very feasible experiments.

Figure 1.

1A: OTUD6B was not the DUB resulting in the most profound growth impairment. 6 DUBs seem to have a more profound effect than OTUD6B on proliferation. What are these and why did the authors decide to investigate OTUD6B? Some discussion is warranted.

1C-D: data show cell cycle and proliferation, but there is no comment regarding extent of apoptosis. Please indicate if cells undergo apoptosis and show proper data (suppl material is fine).

1F: why was the DUB activity assay performed in a distinct cell line than those previously examined. Can the authors perform this experiment in 1-2 MM cell lines?

Was there a reason why the authors decided to use shRNA rather than CRISPR for their functional experiments? As the KO phenotype appears to be of growth arrest rather than lethality, most of the experiments could have been performed using CRISPR-Cas9 gene editing.

Figure 2. Why were in vivo ubiquitylation and pull down of endogenous LIN28B performed in HEK293T cells? Would the authors be able to perform these experiments in at least one MM cell line?

2C. In cells overexpressing the catalytic dead OTUD6B there appears to be ongoing deubiquitylation of LIN28B as compared to non OTUD6B overexpressing cells. How do the authors explain this? Is this the effect of the endogenous WT OTUD6B or do they think that there is redundancy in the deubiquitylation of LIN28B?

Figure 3. Would the authors be able to reproduce these experiments in at least another MM cell line other than MM.1S?

3A: please add an explanation regarding the nature of the double band seen in lane 2 and 4 for LIN28B in WCE.

3B. Similarly, a double band is seen here for OTUD6B. Please explain.

3D. Can these data be shown in 1-2 MM cell lines rather than A459?

Figure 4G: shOTUD6B tumors appear pauci-cellular with significantly less MM cells as compared to the others. Is this an issue with the specific tumor shown or else is there apoptosis occurring? A more representative field/tumor should be shown in the former case.

Figure 5D: Can the author show a rescue experiment for Myc downregulation in shOTUD6B by overexpressing LIN28B?

It would be helpful to add a schema as closing figure, detailing the presumed molecular mechanisms underlying the oncogenic role of OTUD6B in MM.

It would be a good idea to discuss DUBs that have already been studied in MM, such as USP14 and UCHL5 and compare them to OTUD6B in terms of suitability as molecular targets of anti-MM therapy. Also, it is critical to mention that an early phase clinical trial of VLX1570, the first in class DUB inhibitor targeting USP14, in combination with dexamethasone in MM was terminated due to grade 5 toxicity. This can provide the opportunity to discuss anticipated, improved therapeutic index of OTUD6B inhibitors based on expression pattern.

I would tone down this statement: "nominate OTUD6B as a mediator of the MGUS-MM transition". The authors show association, not causality. Something like "a potential mediator" would be more accurate.

Overall, I hope to see the revised manuscript published very soon.

One citation that the authors should consider including is PMID: 19164601, in which the dependency of MM cells on the ubiquitin-proteasome pathway is first described.

Dear Kelly,

We are enclosing a revised version of our manuscript (Paulmann *et al.*, EMBOJ-2022-110871) titled "*The OTUD6B-LIN28B-MYC axis determines the proliferative state in multiple myeloma*" in which we have addressed all specific and general issues raised by the referees in our original submission.

The various revision experiments outlined below strictly follow the detailed revision plan and strategy discussed with you beforehand. We are confident that we have now addressed all the issues raised by the reviewers in full.

Please find below our detailed point-by-point response to the reviewers' comments. Respective changes are highlighted in the new revised manuscript in yellow.

Reviewer #1:

This reviewer acknowledges the interest of our work and states "*This is an interesting study with extensive work.*". He/she however asks that different comments should be addressed to strengthen the manuscript:

Major comments:

1/ *The results presented in the Figure 1B demonstrating the effect on cell cycle have been obtained with different cell lines. However, these cell lines are associated with the same cytogenetic abnormality resulting in MAF deregulation. MAF translocations/deregulations represent the smaller molecular subgroup in MM with 8% of the patients. The authors should validate the biological results of OTUD6B depletion on cell lines from other molecular subgroups reflecting the molecular heterogeneity of the disease.*

To address this important point, we performed proliferation and cell cycle experiments under conditions of sh-RNA mediated knock-down of OTUD6B in the MM cell lines L363 and U266,

which do not contain MAF translocations/deregulations. These experiments demonstrate identical effects of OTUD6B silencing on both proliferation and the cell cycle as observed in the other MM cell lines (new Fig. 1B, C, Appendix Figure S2B). We therefore conclude that the effects of OTUD6B are independent of MAF translocations/deregulations. This is now also indicated in the manuscript on page 5.

“What are the effects of OTUD6B depletion on MM cell survival?”

We addressed this important question by analyzing Caspase 3 cleavage in OTUD6B silenced MM cells and indeed found substantially increased levels of cleaved Caspase 3 (new Fig. Appendix Figure S2C). Likewise, increased Caspase 3 cleavage is also present in the in vivo MM xenograft tumors upon OTUD6B knock down (new Fig. 4C). Thus, the prolonged arrest of MM cells at G1/S is indeed followed up by apoptosis. We indicate this finding on page 5 of the manuscript.

“The authors should validate the reversion of the phenotype with OTUD6B supplementation.”

To address this point, we generated MM cells with doxycycline inducible OTUD6B expression. Within the setting of OTUD6B-knockdown using a 3'UTR-binding shRNA, induced OTUD6B expression indeed rescued the proliferation phenotype to a large part (new Fig. EV1F, G). The lack of complete rescue is likely due to the fact, that we only re-express IF1 of OTUD6B and thus the effects of silencing other IFs of OTUD6B, which are also targeted by the shRNA approach, remain active. Against this background, we would like to argue that these experiments clearly demonstrate that the effect of shRNA mediated knock down of OTUD6B is indeed mediated by the loss of this DUB. This new experimental approach is mentioned on page 5 of the manuscript.

“It would be interesting to know if OTUD6B results or not in DNA damages and DDR.”

Here, we included phospho-gamma H2AX, phospho-ATR phospho-ATM and phospho-p53 staining into the analysis of OTUD6B silenced MM cells and did not observe any signal of DNA damage caused by OTUD6B inactivation (please see Fig. 1 below).

Fig. 1. OTUD6B depletion does not cause DNA-damage.

MM1.S cells were lentivirally infected with the indicated shRNAs and analyzed by immunoblot at day 4 post infection. MM1.S-WT cells treated with Etoposid or Doxorubicin (both 5 μM for 2 hrs) serve as a positive control for DNA damage. Membranes were probed with the indicated antibodies. CUL-1 serves as a loading control.

Additionally, we included p-p53- and p53 blots into Figure 1E, demonstrating that the failure of MM cells to enter S-phase upon OTUD6B depletion does not occur due to DNA-damage induction.

2/ *“The authors identified a downregulation of MYC after OTUD6B or LIN28B depletion. Since the MYC-IRF4 axis plays a major role in MM biology, the authors should investigate the effect on IRF4 expression.”*

We interrogated our RNAseq data set from RPMI 8226 MM cells (sh_Ctrl vs. sh_OTUD6B) and did not detect any changes in IRF4 expression. We therefore concluded not to further follow up on this point, in agreement with the editor.

3/ *“The authors used publicly available cohorts and identified that high OTUD6B expression is associated with a poor outcome. However, the GSE2658 dataset is composed of two different cohorts TT2 and TT3 with differences in the treatment. The authors should investigate the prognostic value in the TT2 and TT3 cohorts separately. Furthermore, the other dataset used GSE24080 comprised the same cohorts TT2 and TT3 than GSE2658.”*

The reviewer is correct and we indeed overlooked this in our previous submission. We have now separately analyzed TT2 and TT3 within GSE24080 and indeed find a significant decrease in the OS of patients with high OTUD6B expression in both cohorts. This is now depicted in the new Fig. 6C.

“The authors should try to identify another independent cohort of patients. The CoMMPASS cohort including newly diagnosed MM patients characterized by RNA sequencing could of particular interest.”

We followed up this very good suggestion of the reviewer and analyzed the CoMMPASS cohort. Indeed, we also find a significant positive correlation between OTUD6B and MYC-target genes (now part of Figure 6B) as well as a significant decrease in the OS of patients with high OTUD6B expression in this cohort, thus further confirming the data obtained from the TT2 and TT3 cohorts (new Fig. 6D).

“In the KM figures, the authors should include the number of patients in each group.”

We now include this information.

“The authors should also investigate the link between OTUD6B expression and the different cytogenetic abnormalities in MM. Since several cytogenetic events are associated with a prognostic value in MM, it will be important.”

We analyzed our internal cohort (n=89) specified in Fig. 6A as to a correlation between OTUD6B expression and major prognostic cytogenetic abnormalities. This analysis revealed no significant correlation (new Fig. EV5). We specify this finding on page 10 of the manuscript.

4/ *“The MYC/IRF4 axis is targeted by IMiDs (immunomodulatory agents) in MM. According to the results presented by the authors, it will be of interest to investigate if OTUD6B depletion could potentiate IMiD MM cell cytotoxicity.”*

As stated above (point#2 of this reviewer), we were not able to identify any changes in IRF4 expression upon OTUD6B knock-down. In agreement with the editor, we therefore decided not to further experimentally follow up this point.

5/ *“The authors presented that OTUD6B depletion potentiates bortezomib toxicity using LP1 cell line. This is not clear why the authors used another cell line compared to the results presented in Figure 1. These results should be validated at least on other MM cell lines. Furthermore, since validation on primary MM cells seems not feasible without specific OTUD6B inhibitor, an in vivo validation using xenograft model would be important.”*

To follow up the suggestion of this reviewer, we performed the respective experiments in another MM cell line (MM1S) and obtained essentially identical results as in the MM LP1 cell line (new Fig. 6H).

As to the validation in the xenograft model, we would like to argue, that such an experiment would likely not be impactful because shOTUD6B xenografted MM tumors hardly grow (see Fig. 4 E, F). An additional effect of bortezomib treatment would therefore not be anticipated to be detectable. In agreement with the editor, we thus did not further follow up on this experimental approach.

“It would be interesting to investigate the response to carfilzomib proteasome inhibitor.”

As suggested, we performed the respective experiment using carfilzomib. Essentially, we find the same results as obtained with bortezomib, although to a somewhat lesser extent (new Fig. 6H).

Reviewer #2:

This reviewer also appreciates our work and states *“... the authors present a largely convincing set of experiments to link OTUD6B with cell cycle-dependent expression of LIN28B...”*.

He/she has different comments that should be addressed prior to publication.

“Although the authors present a largely convincing set of experiments to link OTUD6B with cell cycle-dependent expression of LIN28B, there is no indication throughout as to how many times experiments including western blots supporting several key findings were performed and this should be rectified”.

We now provide this information for all experiments throughout the manuscript.

“In particular, the data suggesting cell cycle-dependent modulation of OTUD6B catalytic activity would benefit from quantitation of multiple replicates (Fig 1F). If reproducible, this would be a striking observation & would merit further investigation.”

We now provide the quantification of this experiment from three independent experiments (new Fig. EV 3D). Please note that the rise of OTUD6B expression at 4 hrs post mitotic release is expected, as these cells move towards the G1/S restriction point, where OTUD6B expression peaks. Cells however release from a nocodazole induced mitotic arrest with variable kinetics which likely explains the variations observed among the 3 independent experiments at this specific timepoint (4 hrs post mitotic release).

Comments:

“Fig 1A: Of the DUB family genes investigated in the CRISPR screen, OTUD6B knockdown lead to

only a comparatively minor survival disadvantage with respect to other targets. It would be interesting to know what these were. Additionally, this minor disadvantage seems to be at odds with the data in Fig 1B, in which OTUD6B knockdown reduced mean proliferation by up to 90% in MMI.S cells”.

The reviewer points towards an issue frequently observed when analyzing the primary data of an CRISPR-based screen. We approached the hits obtained from this screen based on their cellular function, available data on essentiality, novelty as a deubiquitylation substrate function, and most importantly, whether the individual hits and their expression correlate with clinical outcome in MM patients (a list of the scoring hits is now provided in Appendix Table S1 as well as their analysis with regard to clinical outcome in Appendix Fig. S1). This integrative analysis identified OTUD6B as the most promising hit as detailed in the manuscript on pages 4 and 5.

The comparatively minor survival disadvantage also results from the fact that only two of the three OTUD6B sgRNAs scored in this screen. Because Fig. 1A represents the means of all sgRNAs for the individual genes, this must be taken into account when evaluating the individual hits.

Finally, the shRNA mediated knock-down of OTUD6B was indeed substantially higher than the effect observed in the initial screen. We would here again like to argue that not all sgRNA scored in the screen but also that shRNA KD is more acute than sgRNA KO, thus resulting in more sustained cell cycle block at the indicated timepoints.

Together, our choice of selecting OTUD6B was evidence based and proved to be the right decision given our further data that subsequently unambiguously validated OTUD6B as a major vulnerability in MM.

“There are several other DUBs showing a similar drop in representation, with 6 exhibiting a much larger reduction, with two of these showing a similar magnitude to some of the essential gene controls. From this panel alone, OTUD6B doesn’t appear to be the best candidate to take forward from the screen. An explanation as to why OTUD6B was selected over these would be useful.”

Please see my detailed comments above that outline our selection strategy.

“Fig 2B: It is not clear how proliferation was assessed. How many replicates were performed?”

We assume the reviewer refers to Fig. 1B. Cells were counted by the trypan blue exclusion method as specified in the methods section. We also added the information to the figure legends. The 3 datapoints of the individual experiments are included in the experiment.

“Fig 2C: It is not necessary to show the K63-specific blot, given that there is no detectable signal to confirm the antibody worked. This panel can be removed.”

The K63-specific blot has been removed.

“Overexpression of catalytically-inactive OTUD6B also appears to cause a reduction in polyubiquitylated LIN28B, albeit less than WT. This should be discussed. Again, these data would be easier to interpret if some indication of reproducibility were provided.

In the accompanying text, the authors state that OTUD6B (C158A) did not deubiquitylate LIN28B in the same experiment, however there does still appear to be a substantial reduction in the ubiquitin

signal (both anti-HA and anti-K48) relative to those cells not expressing OTUD6B (compare lanes 3 and 5).”

The OTUD6B (C158A) mutant in our hands is mostly inactive but indeed retains some residual activity. Personal communications of different researchers in the field of ubiquitin have indicated similar observations with other cysteine-based DUBs. Based on the reproducible strong effect observed along with the other data demonstrated in the manuscript, we are confident that stabilization of LIN28B at G1/S is indeed dependent on the deubiquitylation activity of OTUD6B. We now also specify the residual activity of the OTUD6B (C158A) on page 6 of the manuscript.

The figure demonstrates exemplary blots from three independent experiments, which is now also indicated in the respective figure legend.

“The conclusions would be stronger if the authors been able to demonstrate accumulation of endogenous LIN28B-pK48 upon 6B knockdown (& subsequent rescue).”

While the point of the reviewer is well taken, it is our experience of many years of research in the ubiquitin field, that fully endogenous ubiquitylation experiments in a mammalian cellular setting are rarely feasible. We have therefore discussed this point with the editor and agreed to not to further follow up on this specific point.

“The LIN28B expressed in cells are all Flag-tagged, a tag which has lysine residues that may contribute to the strong ubiquitylation observed. Complementing the data shown with endogenous protein or identifying lysine residues in LIN28B that are ubiquitylated will address this concern.”

We addressed this question in two experiments. First, we looked at another FLAG-tagged substrate candidate of OTUD6B identified by our MS experiments, which however proved to not be a substrate. In the respective deubiquitylation experiment, no effect of OTU6B expression was observed, arguing that OTUD6B does not deubiquitylate a possible FLAG ubiquitylation. In a second effort, we tested untagged-LIN28B together with HA-Ubiquitin. Here, we also observed deubiquitylation of LIN28B, thus further ruling out the possibility that OTUD6B targets the FLAG tag for LIN28B deubiquitylation. (Please see Figure 2 below for both experiments)

Fig. 2. OTUD6B deubiquitylation activity is not FLAG-tag specific. **A**, *In-vivo* ubiquitylation assay of MTDH in OTUD6B or control vector overexpressing cells. HEK293T cells were transfected with indicated combinations of FLAG-MTDH, HA-Ubiquitin, OTUD6B and EV control and treated with MG132 for 3 hrs 24 hrs later. Denatured WCE were subjected to FLAG-IP. **B**, *In-vivo* ubiquitylation assay of untagged LIN28B in OTUD6B or control vector overexpressing cells. HEK293T cells were transfected with indicated combinations of LIN28B, HA-Ubiquitin, OTUD6B and EV control and treated with MG132 for 3 hrs. Denatured WCE were subjected to HA-IP. WCEs and IPs were analyzed by immunoblotting using the indicated antibodies.

“Fig 2D: It is somewhat surprising that OTUD6B interacts with non-ubiquitylated LIN28B. Does it preferentially immunoprecipitate the polyUb form purified from cells in panel E?”

In this experiment, LIN28B was purified from HEK293 cells and is therefore likely ubiquitylated.

Fig 3B-D: These would benefit from normalised quantitation of LIN28B expression from experimental replicates.

As suggested by the reviewer, we quantified LIN28B in the experiments shown in Figs. 3B-D, that were each performed in triplicates. These quantification data are now shown in Appendix Fig. S4A-C.

“Fig 4B, C: It is not clear whether the individual data points presented represent technical replicates within a single experiment or pooled data from separate experiments. If the former, how many replicates were performed?”

These figures demonstrate the results from three independent biological replicates.

“Fig 4C, D: The equivalent levels of LIN28B in RFP controls of both control and OTUD6B-silenced cells 8h post G1/S release is difficult to reconcile with the data at similar time points in Fig 3D, especially given the extension of G1/S phase in the knocked down cells. This should be addressed.”

Figure 3D marks a half-life experiment that was performed with CHX and thus does not demonstrate steady state expression as demonstrated in Figs 4C, D. Figs 3D and 4C, D are therefore difficult to compare. This point was also discussed with the editor accordingly.

“Additionally, the maintenance of Cyclin E upon OTUD6B knockdown, even during overexpression of LIN28B, suggests the possible existence of additional OTUD6B substrates relevant for cell cycle progression. This data should be quantitated and discussed.”

We now show quantification of Cyclin E in the new Appendix Fig S3E. These data show that exogenous expression of LIN28B indeed largely rescues the OTUD6B phenotype. In principle, we can however not fully exclude the existence of other substrate(s) of OTUD6B, which however does not compromise the conclusions drawn in this manuscript. This discussion is now included on page 12 of the manuscript.

“Fig 4G: The claimed increase in p27 expression in OTUD6B knockdown tumors is not convincing in the image shown. If anything, there appears to be more p27 staining in the control tumor.”

shOTUD6B tumors are largely necrotic and IHC is hard to perform. This is particularly true for the staining with the p27 antibody. We have therefore decided to remove the p27 stains and instead replaced these with cleaved Caspase 3 stains. These new stains show a strong increase in cleaved Caspase 3 upon knock-down of OTUD6B and LIN28B that corresponds to the cell line data shown in Appendix Fig. S2C.

“Fig 6F: The preceding data identify OTUD6B-mediated rescue of LIN28B from proteolytic degradation as a promotor of enhanced proliferation in multiple myeloma cells. These effects are opposed by OTUD6B knockdown and consequent increased LIN28B degradation. Since proteasome inhibition would be expected to rescue LIN28B levels, it is somewhat surprising that the authors

observed synergistic effects of bortezomib and OTUD6B knockdown upon suppression of proliferation. Any explanation or discussion?"

The reviewer is correct with this statement and we ourselves got puzzled over this question when obtaining the data. Our best explanation is that under sublethal doses of bortezomib the proteasome is not fully inhibited, thus allowing for some LIN28B degradation, while the overall proteotoxic stress effect of bortezomib is present and works synergistically with OTUD6B knock-down. This point is now also discussed in the “discussion section” of the manuscript on page 12.

Minor comments:

“- In the figure 5A legend: GESA -> GSEA.”

We have corrected this misspelling.

“- The discussion feels lacking in the context of what is already known about OTUD6B biology, though admittedly there have been few high-quality studies. Mutations in OTUD6B, presumably resulting in a loss of function, are known to result in an intellectual disability disorder with dysmorphic features in humans, and corresponding KO is embryonically lethal in mice (Santiago-Sim et al, 2017 - PMID: 28343629). This is perhaps of relevance with LIN28B as a developmentally regulated protein, and worth considering in the context of OTUD6B inhibition?"

We now discuss this paper within the “discussion section” on pages 11 and 12 of the manuscript.

Referee #3:

This reviewer is very enthusiastic about our work and states: *“This is an exceptionally well-crafted manuscript with scrupulous methodology, high degree of novelty and potential impact. As a reviewer, I have hardly ever read a manuscript and found no major flaws to it. This work was a pleasure to read and I commend the authors for their thorough work.”*

He/she has some minor points:

“Figure 1.

1A: OTUD6B was not the DUB resulting in the most profound growth impairment. 6 DUBs seem to have a more profound effect than OTUD6B on proliferation. What are these and why did the authors decide to investigate OTUD6B? Some discussion is warranted.”

This important question has also been asked by reviewer #2 and we provide extensive information on our strategy towards the selection of OTUD6B above (pages 4 and 5 of this rebuttal letter).

“1C-D: data show cell cycle and proliferation, but there is no comment regarding extent of apoptosis. Please indicate if cells undergo apoptosis and show proper data (suppl material is fine).”

We addressed this important question by analyzing Caspase 3 cleavage in OTUD6B silenced MM cells and indeed found substantially increased levels of cleaved Caspase 3 (new Appendix Fig. S2C). Likewise, increased Caspase 3 cleavage is also present in the in vivo MM xenograft tumors upon

OTUD6B knock down (new Fig. 4C). Thus, the prolonged arrest of MM cells at G1/S is indeed followed up by apoptosis. We indicate this finding on page 5 of the manuscript. Please also see our respective comments to the same question of reviewer #1.

“1F: why was the DUB activity assay performed in a distinct cell line than those previously examined. Can the authors perform this experiment in 1-2 MM cell lines?”

This experiment was indeed performed in A549 (lung adenocarcinoma) cells for technical reasons. Such experiments require a precise cell cycle synchronization of the entire cell population studied which is typically only possible in adherent cells. For this reason, this experiment was performed in A549 cells. Given the comment of this reviewer, we also tried this experiment in RPMI8226 MM cells, which marks the only MM line in our hands, that can be sufficiently synchronized at the G1/S transition as well as in mitosis. Indeed, we also observe the same activity pattern of OTUD6B (peak activity at G1/S) in RPMI8226 (new Fig. EV2E).

“Was there a reason why the authors decided to use shRNA rather than CRISPR for their functional experiments? As the KO phenotype appears to be of growth arrest rather than lethality, most of the experiments could have been performed using CRISPR-Cas9 gene editing.”

Many of the demonstrated experiments require timely and synchronous KD and this, in our hands, is better achieved using shRNAs. We therefore chose this approach in many of the experiments.

“Figure 2. Why were in vivo ubiquitylation and pull down of endogenous LIN28B performed in HEK293T cells? Would the authors be able to perform these experiments in at least one MM cell line?”

With regard to the pull-down experiments of endogenous LIN28B in MM cells, this experiment was already included in the previous manuscript in Fig. 3A (MM1S cells) and was maybe not seen by the reviewer while evaluating Figure 2.

With reference to the in vivo ubiquitylation experiments, such experiments require transfection with multiple constructs (HA-Ubiquitin, LIN28B/EV, OUTD6B/OTUD6B variant/EV) in different combinations. Such experiments are technically hardly feasible in MM cells which require viral transduction for transgene expression and triple viral infections rarely result in equal expression of the individual transgenes within the different experimental conditions studied. We therefore chose to perform these experiments in 293T cells, which is an accepted model for such experiments in the field.

“2C. In cells overexpressing the catalytic dead OTUD6B there appears to be ongoing deubiquitylation of LIN28B as compared to non OTUD6B overexpressing cells. How do the authors explain this? Is this the effect of the endogenous WT OTUD6B or do they think that there is redundancy in the deubiquitylation of LIN28B?”

This point was also raised by Reviewer 2 and we therefore would like refer to page 6 of this rebuttal letter. Briefly, the OTUD6B (C158A) mutant in our hands is mostly inactive but indeed retains some residual activity. Personal communications of different researchers in the field of ubiquitin have indicated similar observations with other cysteine-based DUBs. Based on the reproducible strong effect observed along with the other data demonstrated in the manuscript, we are confident that

stabilization of LIN28B at G1/S is indeed dependent on the deubiquitylation activity of OTUD6B. We now also specify the residual activity of the OTUD6B (C158A) on page 6 of the manuscript. However, we agree with the reviewer that endogenous OTUD6B activity may also be involved and maybe there is a further unknown DUB of LIN28B that we are not aware of. Investigating this question however is beyond the scope of this manuscript and given our data, we are confident that this observation does not compromise our conclusions.

“Figure 3. Would the authors be able to reproduce these experiments in at least another MM cell line other than MM.1S?”

MM cells are typically very difficult to synchronize. In our hands, we can synchronize and release RPMI8226 cells at G1/S and mitosis to an acceptable extent while MM1S cells only synchronize at G1/S (please also see our comment to the point re. Fig. 1F of this reviewer). In the previous submission, Fig. 3B was mistakenly labeled to be performed in MM1S while it was performed in RPMI8226 for reasons outlined above. This has been corrected in the current manuscript. At the same time, we were not able to perform dual cell cycle profiles (G1S & mitosis) in MM1S cells for which reason we provide data in well synchronizable adherent A549 cells, that well reproduce our finding in RPMI8226 cells. This strategy was also beforehand discussed with the editor.

“3A: please add an explanation regarding the nature of the double band seen in lane 2 and 4 for LIN28B in WCE”

The upper part of this double band represents a phosphorylated form of LIN28B, of so far unknown significance, which we were able to specify by phosphatase treatment (please see Fig. 3 below). The phosphorylation event appears to occur during G2/M and is only visible upon long exposures of immunoblots. Further investigations of this phosphorylation event are beyond the scope of this manuscript.

Figure 3: LIN28B is phosphorylated during mitosis. HEK293T cells were treated with nocodazole for 24 hrs and the resulting WCE split in half. One half was treated with Lambda-Phosphatase and analyzed by immunoblot alongside the untreated sample using the indicated antibodies.

“3B. Similarly, a double band is seen here for OTUD6B. Please explain.”

We believe that the lower part of this double band represents a so far undescribed isoform of OTUD6B which seems typical for RPMI8226 and KMS12BM MM cells (Fig. 5C and Fig. EV1E, Appendix S1B, EV2E), highlighting once more, that it is still unclear how many isoforms of OTUD6B exist and if they fulfil different functions. Isoform 1 of OTUD6B (UniProt (Q8N6M0-1)) was used for overexpression studies, as it seems to be the one expressed throughout all MM lines tested. While we can exclude phosphorylation from phosphatase treatment experiments, another post-translational modification of OTUD6B may also be possible.

“3D. Can these data be shown in 1-2 MM cell lines rather than A459?”

We extensively tried to perform this experiment in the MM cell lines MM1S and RPMI8226. However, the combination of viral transduction, G1/S synchronization **and** CHX treatment resulted in substantial apoptosis of these MM cells. We therefore need to rely on adherent cells (such as A549) for this particular type of experiment.

“Figure 4G: shOTUD6B tumors appear pauci-cellular with significantly less MM cells as compared to the others. Is this an issue with the specific tumor shown or else is there apoptosis occurring? A more representative field/tumor should be shown in the former case.”

shOTUD6B tumors are largely necrotic/apoptotic and IHC is hard to perform. This is particularly true for the staining with the p27 antibody. We have therefore decided to remove the p27 stains and instead replaced these with cleaved Caspase 3 stains. These new stains show a strong increase in cleaved Caspase 3 upon knock-down of OTUD6B and LIN28B indicative of an induction of programmed cell death.

“Figure 5D: Can the author show a rescue experiment for Myc downregulation in shOTUD6B by overexpressing LIN28B?”

We performed such an experiment in RPMI8226 MM cells which express LIN28B under the control of a doxycycline inducible promoter. Upon simultaneous OTUD6B depletion and induced LIN28B expression, we could partially but significantly restore MYC levels. These data are now included in the new Appendix Fig. S45D.

“It would be helpful to add a schema as closing figure, detailing the presumed molecular mechanisms underlying the oncogenic role of OTUD6B in MM.”

We will include such a scheme within the synopsis format of this journal.

“It would be a good idea to discuss DUBs that have already been studied in MM, such as USP14 and UCHL5 and compare them to OTUD6B in terms of suitability as molecular targets of anti-MM therapy. Also, it is critical to mention that an early phase clinical trial of VLX1570, the first in class DUB inhibitor targeting USP14, in combination with dexamethasone in MM was terminated due to grade 5 toxicity. This can provide the opportunity to discuss anticipated, improved therapeutic index of OTUD6B inhibitors based on expression pattern.”

We have incorporated these points into the discussion (pages 11 and 12).

“I would tone down this statement: “nominate OTUD6B as a mediator of the MGUS-MM transition”. The authors show association, not causality. Something like “a potential mediator” would be more accurate.”

We have corrected accordingly.

We sincerely appreciate the constructive suggestions made by the reviewers and strongly believe that this revision process has improved the clarity and message of the paper.

Klinikum rechts der Isar

Technische Universität München

We are looking forward to hearing from you.

On behalf of all authors,
Sincerely,

Florian Bassermann M.D./Ph.D.

Dear Florian,

Congratulations on a great revision! Overall, the referees and I feel you have satisfied the majority of concerns and I thank you for your comprehensive response to the referee comments. However, there remain a few editorial items to address in an updated manuscript as follows:

- The "Data Availability" section should directly follow the Methods section.
- Thank you for providing a synopsis image. Along with this image, please provide me with a general summary statement and 3-5 bullet points that capture the key findings of the paper (for examples, see: <http://emboj.emboress.org/>)
- We strongly encourage the publication of source data, particularly for electrophoretic gels and blots and graphs, with the aim of making primary data more accessible and transparent to the reader. It would be great if you could provide me with a PDF file per figure that contains the original, uncropped, and unprocessed scans of all or key gels used in the figure or for graphs, an Excel spreadsheet with the original data used to generate the graphs. The PDF files should be labeled with the appropriate figure/panel number and should have molecular weight markers; further annotation could be useful but not essential. The PDF files will be published online with the article as supplementary "Source Data" files.
- On page 5, second paragraph the word homozygous is misspelled, please correct.

I look forward to receiving your revised version soon and to moving forward with this manuscript.

All the best,
Kelly

Kelly M Anderson, PhD
Editor
The EMBO Journal
k.anderson@embojournal.org

Further information is available in our Guide For Authors: <https://www.emboress.org/page/journal/14602075/authorguide>

Use the link below to submit your revision:

Link Not Available

Referee #2:

These revisions & rebuttal satisfy most of my concerns. It would be good to know the mechanism controlling cell cycle-specific modulation of 6B activity but not essential for publication.

The authors' explanation that the C158A 6B mutant retains some residual activity (rebuttal p5/6) is not especially convincing but at least it is now acknowledged.

They also appear to have misunderstood my query re: surprising ability of 6B to IP non-ubiquitylated LIN28B & whether it would preferentially pull down the polyUb'd form (top p7 rebuttal).

Nevertheless, I am happy for this manuscript to be accepted for publication.

Referee #3:

Dear Dr. Anderson,
the authors carefully addressed all the points of the reviewers in a convincing manner. When experiments that were suggested were not performed, a suitable explanation was provided.

This is a great paper and I look forward to seeing it published.

The authors performed the requested editorial changes.

Dear Florian,

Congratulations on an excellent manuscript, I am pleased to inform you that your manuscript has been accepted for publication in the EMBO Journal. Thank you for attending to the final editorial concerns in the latest version.

I will now begin the final checks before submitting to the publisher next week. Once at the publisher, it will take about 3 weeks for your manuscript to be published online. As a reminder, the entire peer review process including referee concerns and your point-by-point response will be available to readers.

It has been a pleasure to work with you to get to the acceptance stage. I will be in touch throughout the final editorial process until publication. In the meantime, I hope you find time to celebrate!

Yours sincerely,

Kelly

Kelly M Anderson, PhD
Editor
The EMBO Journal
k.anderson@embojournal.org

Please note that it is EMBO Journal policy for the transcript of the editorial process (containing referee reports and your response letter) to be published as an online supplement to each paper. If you do NOT want this, you will need to inform the Editorial Office via email immediately. More information is available here:
<https://www.embopress.org/page/journal/14602075/authorguide#transparentprocess>

Your manuscript will be processed for publication in the journal by EMBO Press. Manuscripts in the PDF and electronic editions of The EMBO Journal will be copy edited, and you will be provided with page proofs prior to publication. Please note that supplementary information is not included in the proofs.

You will be contacted by Wiley Author Services to complete licensing and payment information. The required 'Page Charges Authorization Form' is available here: https://www.embopress.org/pb-assets/embo-site/tej_apc.pdf - please download and complete the form and return to embopressproduction@wiley.com

Should you be planning a Press Release on your article, please get in contact with embojournal@wiley.com as early as possible, in order to coordinate publication and release dates.

If you have any questions, please do not hesitate to call or email the Editorial Office. Thank you for your contribution to The EMBO Journal.